# Aerobactin is a key driver of hypervirulent *Klebsiella pneumoniae* translocation and virulence

**Giovanna E. Hernandez[1], Juan D. Valencia-Bacca[1], Emma F. Bennett[1], Md. Maidul Islam[2], Suhrid Maiti[2], Noah A. Nutter[1], Taylor M. Young[1], David L. Caudell[3], Alicia Costa-Terryll[1], Jaden J. Skelly[1], Edison T. Floyd[4], Hannah M. Atkins[5], M. Ammar Zafar**[1,2]*

**1** Department of Microbiology and Immunology, Wake Forest School of Medicine, Winston Salem, North Carolina, United States of America, **2** Department of Microbiology and Immunology, Emory University School of Medicine, Atlanta, Georgia, United States of America, **3** Department of Pathology-Comparative Medicine, Wake Forest School of Medicine, Winston-Salem, North Carolina, United States of America, **4** Pathology Services Core, University of North Carolina Chapel Hill, Chapel Hill, North Carolina, United States of America, **5** Division of Comparative Medicine, Department of Pathology and Laboratory Medicine, University of North Carolina Chapel Hill, Chapel Hill, North Carolina, United States of America

* ammar.zafar@emory.edu

## Abstract

Hypervirulent *Klebsiella pneumoniae* (hvKP) is an emerging pathotype capable of causing severe systemic infections. Gastrointestinal (GI) colonization often precedes invasive disease, but the mechanisms driving translocation to extra-intestinal sites are not known. A hallmark of hvKP is a large virulence plasmid encoding a number of virulence factors, including aerobactin, a siderophore associated with enhanced virulence. Using a murine GI colonization model with an intact microbiota, we examined if aerobactin enabled hvKP to translocate, by comparing an hvKP clinical isolate (hvKP1) to an isogenic aerobactin biosynthesis mutant (*iucA*-). Both strains colonized the GI tract similarly, but mice colonized with the *iucA*- mutant exhibited significantly lower bacterial burdens in the extra-intestinal organs, indicating a defect in translocation. This defect was not observed in a systemic infection model, suggesting a specific role for aerobactin in GI translocation. In cell culture based assays, the *iucA*-mutant showed reduced adhesion, invasion, and translocation. Both *in vivo* and *in vitro* data supported a transcellular route of translocation. Notably, the *iucA*- mutant displayed increased hypermucoviscosity (HMV) linked to the upregulation of the *rmp* locus, likely impairing host cell adhesion. These findings demonstrate that tight regulation of HMV through metal homeostasis mediated by aerobactin, promotes hvKP translocation across the intestinal epithelium by enhancing adhesion and cellular entry. This work offers new insight into hvKP pathogenesis and informs potential strategies to limit its invasive potential.

**Data availability statement:** All data supporting the findings of this study are available within the manuscript and its Supporting information files.

**Funding:** MAZ and GEH were supported by R01AI173244 and R21AI166642, Division of Intramural Research, National Institute of Allergy and Infectious Diseases. GEH was supported by AI173244S1 and by the Training Progam in Immunology and Pathogenesis grant T32AI007401 awarded to Wake Forest School of Medicine, Division of Intramural Research, National Institute of Allergy and Infectious Diseases. GEH was also supported through the Burroughs Wellcome Fund graduate diversity enrichment program. The Pathology Services Core (PSC, UNC-Chapel Hill) is supported in part by P30CA016086, Division of Intramural Research, National Cancer Institute, and by a North Carolina Biotech Grant 2024-IIG-0015, awarded to the University of North Carolina at Chapel Hill. The funders had no role in study design, data collection and analysis, decision to publish, or preparation of the manuscript.

**Competing interests:** The authors have declared that no competing interests exist.

## Author summary

The emergence of hypervirulent *Klebsiella pneumoniae* (hvKP) poses a significant public health threat due to its capacity to cause severe, often fatal, systemic infections. Of further concern are convergent isolates that are multidrug-resistant and harbor virulence factors, making them difficult to treat. Although gastrointestinal (GI) colonization often precedes invasive disease, the molecular mechanisms driving hvKP translocation from the GI tract to peripheral organs remain unclear. Here, we identify a previously uncharacterized role for the siderophore aerobactin in promoting translocation by modulating the bacterium's hypermucoid capsule, thereby enhancing its ability to adhere to and traverse host cells. These findings advance our understanding of how hvKP transitions from a gut colonizer to invasive pathogen and suggest new avenues for preventing invasive disease.

## Introduction

The gastrointestinal (GI) mucosa serves as a critical barrier between the body and the external environment, and its integrity is essential for maintaining a healthy gut. When microbes overcome this barrier, bacterial translocation, defined as the passage of viable bacteria from the GI tract to systemic sites, can lead to severe invasive disease [1,2]. Three principal mechanisms are thought to promote translocation: bacterial overgrowth, epithelial barrier disruption, and impaired host immunity [2]. Enteric pathogens have developed strategies to facilitate gut colonization and systemic dissemination. For instance, *Salmonella*, *Yersinia*, *Shigella,* and *Listeria* species colonize the GI tract and utilize a transcellular (intracellular) route of translocation to breach the epithelium [3], whereas *Vibrio cholerae* and *Campylobacter jejuni* disrupt tight junction proteins to traverse via a paracellular (intercellular) route [4,5].

*Klebsiella pneumoniae* (*Kpn*), a gram-negative enteric pathobiont, is a leading cause of hospital-acquired infections. Epidemiological studies have established that *Kpn* gut carriage is often associated with systemic disease, effectively making the GI tract a key reservoir [6–8]. While *Kpn* is regarded as an opportunistic pathogen, primarily causing infection in immunocompromised hosts, the emergence of more virulent isolates associated with severe community-acquired infections has led to the classification of two pathotypes: classical and hypervirulent [9]. Hypervirulent (hvKP) isolates exhibit a higher pathogenic potential due to the inclusion of a large virulence plasmid that encodes an arsenal of genetic factors, leading to the formation of a hypercapsule, hypermucoviscosity (HMV), and the production of additional siderophores [9]. The virulence plasmid contains the *rmp* locus, genes within it are known to positively regulate capsule (*rmpC*) and the HMV phenotype (*rmpD*), which is characterized by more uniform capsular polysaccharide chain lengths [10]. Additionally, siderophore loci encoding for salmochelin (*iro*) and aerobactin (*iuc*) are also located on the virulence plasmid [9]. These virulence factors augment the ability of hvKP to

evade the immune system, including overcoming complement mediated clearance [11], innate immune protein Lipocalin 2 mediated iron sequestration [12], and modulate the host Rho GTPase and phosphatidylinositol 3-kinase (PI3K)/Akt cell signaling to trigger cytoskeleton rearrangement, promoting hvKP translocation [13]. While mechanistic studies have revealed the involvement of host and bacterial factors in the translocation of enteric pathogens [14,15], the underlying mechanisms remain poorly defined and require further investigation.

Siderophores are key virulence determinants, enabling the bacteria to acquire iron in iron-limited environments, such as those found within the host. Siderophores are iron-chelating molecules with a high affinity for ferric iron ($Fe^{3+}$), which they bind and transport back to the cell where it is reduced to ferrous iron ($Fe^{2+}$) for cellular processes [16]. Hypervirulent strains generally produce four siderophores: enterobactin, yersiniabactin, salmochelin, and aerobactin, with aerobactin accounting for over 90% of total siderophore production under specific growth conditions [17]. Genes encoding aerobactin biosynthesis have also been identified in other enteric pathogens, including *E. coli*, *Vibrio*, *Salmonella*, and *Shigella* species [18]. In co-culture, *Vibrio* strains that produced aerobactin maintained a competitive advantage over strains lacking the aerobactin uptake system, effectively restricting their growth [19]. In *Shigella*, aerobactin enhances virulence and is vital for extracellular growth; however, expression of the aerobactin biosynthesis genes is repressed during intracellular growth, even when the bacteria are starved for iron [20].

Although the contribution of aerobactin to hvKP virulence has been well established [17,21], its role in intestinal colonization and dissemination has not been directly explored. Prior studies have primarily focused on the function of siderophores in *Kpn* models of lung, systemic, or subcutaneous infection [17,21–24], overlooking their potential roles in GI colonization and translocation. In this study, we investigated the role of aerobactin in hvKP gut colonization in the presence of an intact microbiota and its role in facilitating systemic dissemination. We demonstrate that while aerobactin is not necessary for gut colonization, it contributes to the translocation process across the intestinal barrier to cause invasive disease. Using a human intestinal cell culture model, we provided further molecular insights into the translocation process and identified the iron-mediated regulation of the mucoid phenotype as being crucial for translocation. These findings establish aerobactin as a key bacterial determinant for hvKP dissemination from the gut to systemic sites, promoting invasive disease.

## Results

### HvKP expresses aerobactin in the murine gut but it is dispensable for gut colonization

Previous studies have demonstrated that of the four siderophores expressed by hvKP strains, aerobactin is the most abundant and a critical virulence determinant [17,21]. The aerobactin operon is well-conserved in hvKP isolates and encodes four biosynthetic genes (*iucA-D*) and the outer membrane receptor (*iutA*) [25,26] (Fig 1A). The hvKP1 clinical isolate (ST86, KL2) harbors the ~200 Kb virulence plasmid pVir/pLVPK, which includes the aerobactin (*iuc1*) gene cluster [27]. Since siderophores play a key role in aiding bacteria to acquiring iron in iron-scarce environments [17,22,28], we examined the growth kinetics of an aerobactin biosynthesis mutant, *iucA::kan* (*iucA-*) in the hvKP1 background, under both iron-replete (LB) and iron-depleted (LB with 200 µM 2, 2'-Dipyridyl [DIP]) conditions. Compared to the wild-type (WT), the *iucA-* mutant had a modest but consistent growth defect in iron-depleted media, whereas no growth defect was observed in iron-replete media (S1A Fig). In competitive co-culture experiments, neither strain outcompeted the other under either growth condition (S1B Fig), suggesting trans-complementation of the *iucA-* mutant by WT, likely due to the presence of an intact *iutA* gene encoding the aerobactin outer membrane receptor. Our *in vitro* qRT-PCR results, comparing RNA isolated from iron-replete and iron-depleted medium, demonstrated that aerobactin gene expression is highly upregulated in the WT strain, more so than the other siderophores (S2A Fig). Loss of aerobactin did not result in an increase in expression of other siderophores except for salmochelin (*iroB*) expression displaying a significant 4-fold increase, relative to the WT. Nevertheless, total siderophore output was significantly lower in the *iucA-* mutant, as measured using chrome azurol S (CAS) agar (S2B Fig).

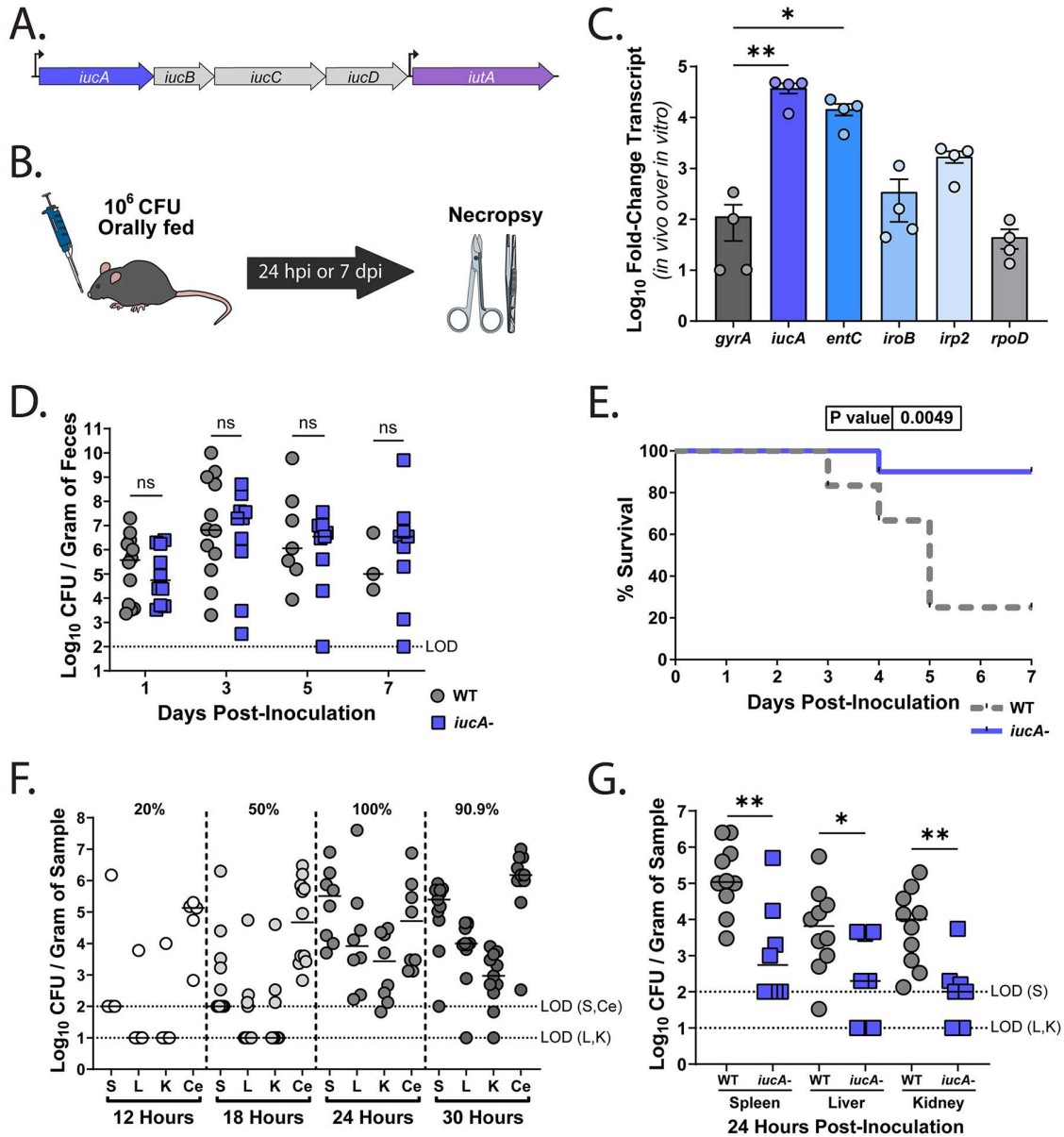

**Fig 1. Aerobactin is dispensable for gastrointestinal colonization but required for efficient translocation and high virulence.** (A) Genetic organization of the *iuc* operon, encoding aerobactin synthesis and receptor genes, in the hypervirulent *K. pneumoniae* clinical isolate hvKP1. The *iucA* gene is shown in blue and *iutA* (receptor) in purple. (B) Schematic of the murine oral feeding model of *Kpn* gastrointestinal colonization and the experimental timeline. (C) RNA was isolated from cecal contents of C57BL6/J mice (n = 4) 4-6 days post-inoculation (PI). The $\log_{10}$ fold-change in transcripts was measured via qRT-PCR for *iucA* (aerobactin), *entC* (enterobactin), *iroB* (salmochelin), and *irp2* (yersiniabactin) and plotted to compare hvKP1 siderophore gene expression in the murine gut (*in vivo*) to in LB media (*in vitro*). Expression of *gyrA* and *rpoD* served as controls, and all data was normalized to the *K. pneumoniae* 16S housekeeping gene for $2^{-\Delta\Delta C_T}$ analysis. Each biological replicate (n = 4) was run in duplicate, and their averages with SEM is shown. Kruskal-Wallis test, followed by Dunn's test of multiple comparisons compared to gyrA control, was performed. (D-E) C57BL6/J mice were orally inoculated with either the WT (gray) or the isogenic *iucA-* mutant (blue) and followed for 7 days (n ≥ 10 per group). (D) Fecal shedding was plotted with each symbol representing a single mouse on the indicated day and the bars specifying the median bacterial shedding. The dashed line represents the limit of detection (LOD). (E) Kaplan-Meier curves compare mice survival between the WT (grey) and the isogenic *iucA-* mutant (blue) inoculated mice (n ≥ 10 per group). Log-rank (Mantel-Cox) tests were performed to compare survival odds. LOD, limit of detection. (F) Translocation kinetics of mice orally fed $10^6$ CFU of the WT strain. Shown is the bacterial burden in the spleen (S), liver (L), kidneys (K), and cecum (Ce) at 12 hours (n = 5), 18 hours (n = 12), 24 hours (n = 8), and 30 hours (n = 11) PI. The percentage of mice with bacteria detected in the extra-intestinal organs (S, L, K), suggesting translocation, is listed at the top of the graph for each time point. The LODs for the samples are depicted as dotted lines, and the pertaining organs are indicated. (G)

Bacterial burden in the extra-intestinal organs of mice at 24 hours PI with either the WT (n = 10) or the *iucA-* mutant (n = 8). (D, G) Mann-Whitney *U* tests were used to determine statistical significance. LOD, limit of detection. *$P < 0.05$, **$P < 0.01$, ***$P < 0.001$, and ****$P < 0.0001$; ns, not significant.

Given the role of the GI tract as a reservoir for *Kpn* [6,7,29], and a transposon mutagenesis screen with an intact microbiome implicating siderophores as important gut colonization determinants [30], we investigated whether hvKP1 expresses siderophores during gut colonization. Using our oral-feeding mouse model of *Kpn* gastrointestinal colonization [29] (Fig 1B), we harvested the cecal contents of mice post-inoculation (PI) with hvKP1 and isolated bacterial RNA. Compared to RNA isolated from broth-grown cultures, qRT-PCR of cecal RNA samples revealed elevated levels of all four siderophores, with aerobactin and enterobactin showing significant upregulation (Fig 1C). These findings establish that there is likely competition for iron in the gut causing upregulation of aerobactin by hvKP residing in the murine gut.

Next, as siderophores have been demonstrated to be important for gut colonization for enteric bacteria during inflammation and antibiotic treatment [31–33], and as aerobactin is expressed by hvKP1 in the mouse GI tract, we tested the *iucA-* mutant in our mouse model and determined the ability of this mutant strain to colonize the murine GI tract. As fecal shedding correlates with density of *Kpn* in the gastrointestinal tract [29], shedding was used as a non-invasive metric of determining gut colonization. Over the course of the 7-day study, both the WT and the *iucA-* mutant shed at equivalent levels (Fig 1D), indicating that aerobactin synthesis is not required for colonization. Interestingly, compared to the WT, mice inoculated with the *iucA-* mutant displayed reduced mortality, displaying a 70% increase in survival (Fig 1E).

Thus, our findings implicate aerobactin in promoting growth under iron-limited conditions and show that, while it is actively expressed in the gut, it is dispensable for GI colonization but contributes to hvKP virulence.

## Aerobactin contributes to translocation of hvKP1 from the gut

Previously, we observed that mice that succumbed to infection following GI inoculation with hvKP1 harbored high bacterial burdens in the liver, spleen, and kidneys [29]. This suggested that bacterial translocation had occurred. To better understand the translocation process, we monitored hvKP1 translocation kinetics by quantifying bacterial loads in the gut and peripheral organs (spleen, liver, kidneys) from 12 to 30 hours PI. At 12 hours PI, hvKP1 was primarily confined to the GI tract. By 18 hours PI, bacteria were detectable in extra-intestinal organs in approximately half the mice, and by 24 hours PI, hvKP1 was recovered from the spleen, liver, and kidneys of all colonized mice (Fig 1F). Given that the *iucA-* mutant colonized the gut similarly to WT (Fig 1D) but exhibited reduced virulence (Fig 1E), we hypothesized that its attenuation was likely due to an impaired ability to translocate from the gut to extra-intestinal sites. Consequently, we selected 24 hours PI to compare bacterial burden at sterile sites between the WT and the *iucA-* mutant. At 24 hours PI, we observed reduced burden of the *iucA-* mutant compared to the WT isolate, in the spleen, liver, and kidneys (Fig 1G). These results identify a potential role for aerobactin in the translocation process.

As the WT strain complemented the growth defect of the *iucA-* mutant under iron-poor growth conditions (S1B Fig), we envisaged the WT strain compensating for the *in vivo* translocation defect of the *iucA-* mutant. We inoculated mice at a 1:1 ratio of the WT and the mutant strain, while both strains colonized the GI tract, the WT maintained a modest competitive advantage. Specifically, the competitive index (CI) in the cecum and colon leaned toward negative values, with mean offsets of -0.14 and -0.19, respectively, suggesting the mutant was slightly outcompeted even in the gut. However, this advantage was more pronounced in the extra-intestinal sites, where the WT strain outcompeted the aerobactin mutant in 4 out of 5 mice, rather than complementing it (Fig 2A). We speculate that, compared to the gut, the differences observed in the systemic sites could be due to the infection barriers present within the host. Additionally, since bacterial burdens varied among WT-colonized mice in both the gut and extra-intestinal sites, we considered whether a correlation existed between the bacterial loads in the gastrointestinal tract and those in the extra-intestinal organs post-translocation. At 24 hours PI, we identified a strong positive correlation between the liver and the cecum, followed by the spleen and the

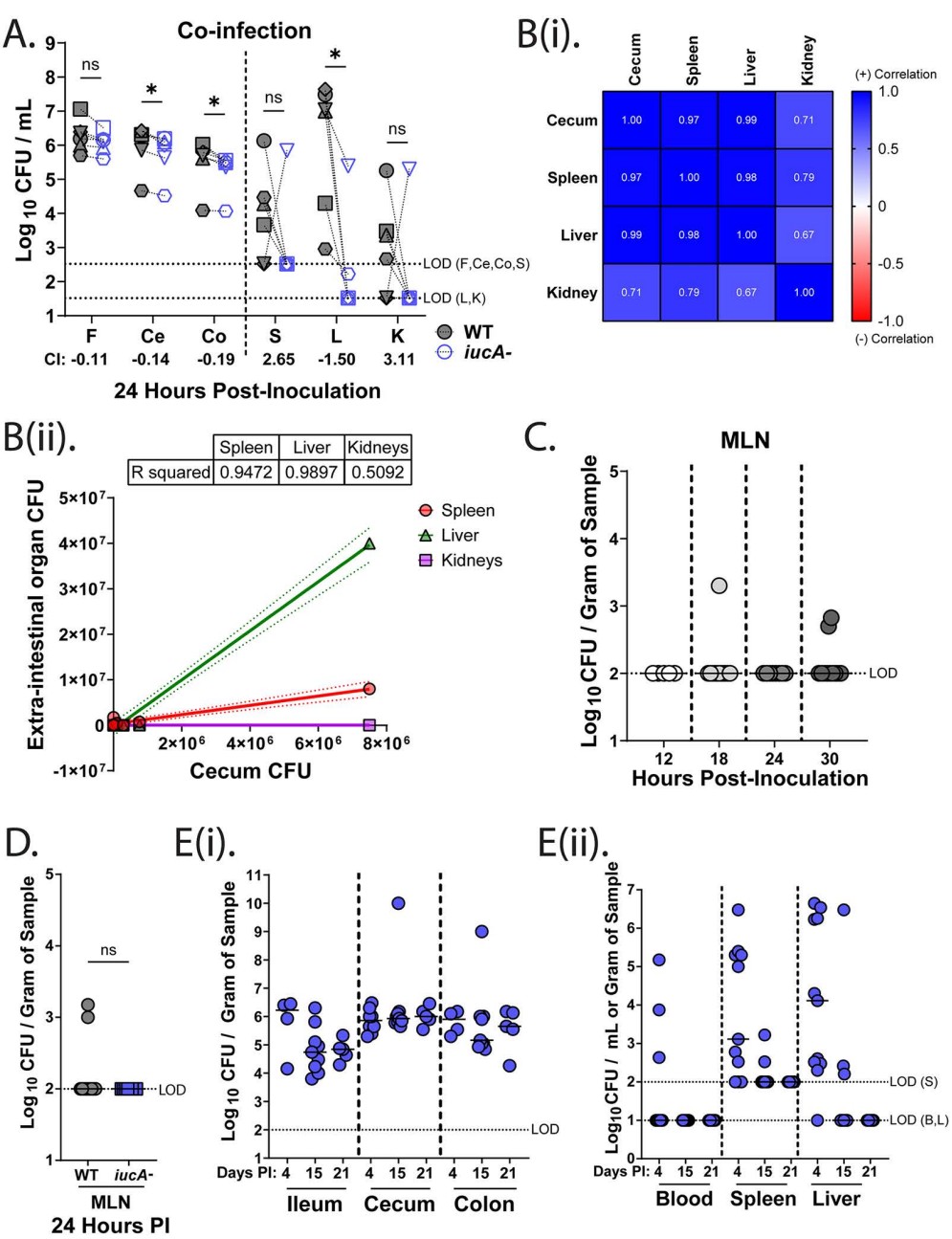

**Fig 2. hvKP dissemination from the gut to systemic sites requires high bacterial burden and does not rely on the lymphatic system.** (A) Competition assay in which mice (n=6) were orally fed a 1:1 mixture of the WT (gray-filled shapes) and the *iucA-* mutant (blue open shapes); each mouse is represented by a unique symbol. Bacterial burden was enumerated from the gastrointestinal samples (F: fecal, CE: cecum, CO: colon) and the extra-intestinal organs (S: spleen, K: kidneys, L: liver) 24 hours post infection (PI). A dotted line connects bacterial densities recovered from the same animal for the specified sample. Competitive index (CI) values were calculated as the $\log_{10}$ average CI for each sample. The dotted lines depict the LOD for the respective samples. (B) Correlation analyses of extra-intestinal (spleen, liver, and kidneys) bacterial burden of hvKP1 relative to gastrointestinal (cecum) colonization at 24 hours PI, displayed as a correlation matrix using the Pearson correlation coefficients (r) (i) and as simple linear regressions (ii). R squared for each regression is listed. (C) CFU obtained from the mesenteric lymph nodes (MLN) of mice orally inoculated with the WT and enumerated at 12 hours (n=5), 18 hours (n=12), 24 hours (n=8), or 30 hours (n=11) PI. A dotted line marks the LOD. (D) Bacterial densities in the MLN at 24 hours PI in mice orally fed either the WT (gray circles) or the *iucA-* mutant (blue squares). The dotted line indicates the LOD. (E) Colonization density of the *iucA-* strain in the (i) gastrointestinal tract (ileum, cecum, and colon) and (ii) the extra-intestinal sites (blood, spleen, and liver) of mice at 4 days, 15 days, and 21 days PI. Statistical significances of CIs were calculated using Wilcoxon signed-rank test with a theoretical median of 0. PI, post-inoculation. *$P < 0.05$, ns, not significant.

cecum (Fig 2B), suggesting gut bacterial burden impacts translocation. These results are in line with a recent *Kpn* molecular study that established that bacterial burden in the lung can impact metastatic spread to secondary sites [34].

Since several enteric pathogens, including *Salmonella enterica serovar Typhimurium*, *Shigella* spp., and *Listeria monocytogenes*, exploit M cells for systemic dissemination [3], we examined the mesenteric lymph nodes (MLNs) in hvKP1-colonized mice. In contrast to reports with *S. Typhimurium* [35], hvKP1 was rarely detected in the MLNs, despite being present in the extra-intestinal organs (Figs 1F, 1G, 2C-2D), suggesting a vascular route for translocation. Furthermore, as mice colonized with *iucA*- still exhibited translocation, with the majority of mice surviving to day 7 (Fig 1E), we wondered whether the surviving mice remained colonized in extra-intestinal sites or cleared *iucA*- over time. We followed mice for either 4, 15, or 21 days PI and observed that the *iucA*- mutant stably colonized the gut through to the study endpoint at 21 days (Fig 2Ei). In contrast, colonization of the extra-intestinal sites declined over time, and by day 21, all mice had cleared the bacteria from systemic sites (Fig 2Eii).

Next, we evaluated the significance of aerobactin in context with the other siderophores produced by hvKP1. We utilized an isogenic strain that only produces aerobactin, *entB::cam irp2::hyg,* and a strain that lacks all four siderophores, Δ*iucA entB::cam irp2::kan*. The aerobactin-only strain colonized the murine gut similarly to WT, whereas the siderophore-null strain had a defect in colonization (Fig 3A), indicating that siderophores contribute to robust gut colonization. Interestingly, the siderophore-deficient mutant was completely avirulent in our murine model (Fig 3B), whereas the aerobactin-only mutant was partially virulent when compared to the WT strain, implying the importance of all four siderophores to hvKP virulence. Collectively, our results demonstrate that hvKP can efficiently translocate from the GI tract to extra-intestinal sites, causing invasive disease, with aerobactin contributing to this process and influencing virulence.

Having observed a difference in bacterial burden at systemic sites post-GI colonization (Fig 1G), we speculated that this difference could either be due to a defect in translocation, or poor survival of the *iucA*- strain in the extra-intestinal space. A recent transposon mutagenesis screen determined *Kpn* bacterial factors important for bacteremia and identified several TonB-dependent iron uptake systems, but none of the siderophore systems were required for colonization of the spleen after systemic infection [36]. Thus, we also used a systemic infection mouse model, in which $10^6$ CFU were injected directly into the bloodstream via the lateral tail vein, bypassing the need for translocation (Fig 4A). No discernable differences in bacterial burden were observed in the systemic sites (spleen, blood, liver, and kidney) between the WT and *iucA*- strains at either 6 or 24 hours PI (Fig 4B, 4C). Furthermore, similar to the survival kinetics post-gut colonization (Fig 1E), we observed reduced mortality in mice orally inoculated with the *iucA*- mutant compared to the WT strain (Fig 4D), providing further evidence that aerobactin impacts hvKP virulence. To further probe whether the *iucA*- mutant had a survival or growth defect in the systemic sites, we evaluated how well it survived in the presence of 10% rabbit serum. Unlike a capsule-deficient mutant, KPPR1S Δ*wcaJ* [37], *iucA*- grew similar to WT in rabbit serum (Fig 4E). These results likely preclude early clearance of the *iucA*- as the reason for reduced burden in the extra-intestinal sites, and instead raise the possibility that the reduced bacterial burden at systemic sites is due to a translocation defect.

## Aerobactin-mediated virulence is conserved across genetically diverse hvKP isolates

Having established that aerobactin is critical for translocation and virulence in hvKP1, we next investigated whether this function is conserved across genetically diverse hvKP strains. *Kpn* exhibits substantial genetic heterogeneity, with over 130 distinct capsule types circulating across classical and hypervirulent pathotypes [38,39]. The hvKP1 isolate is a K2 capsule type, and since the most prevalent capsule types for hvKP strains are K1 and K2 [40], we examined additional clinical isolates, including hvKP2 (ST23, KL1) and hvKP94 (ST23, KL1), along with their isogenic aerobactin-deficient counterparts (*iucA::kan*). We observed no difference in fecal shedding from mice colonized with hvKP2 or hvKP2 *iucA*- (Fig 5A). The hvKP94 isolate is considered as a partially virulent strain [23], and hvKP94 *iucA*- also behaved similarly in the gut (Fig 5B). In accordance with other murine challenge studies [23], compared to hvKP2, the partially virulent strain hvKP94, in our murine model of infection had reduced virulence. Conversely, both mutant strains presented reduced

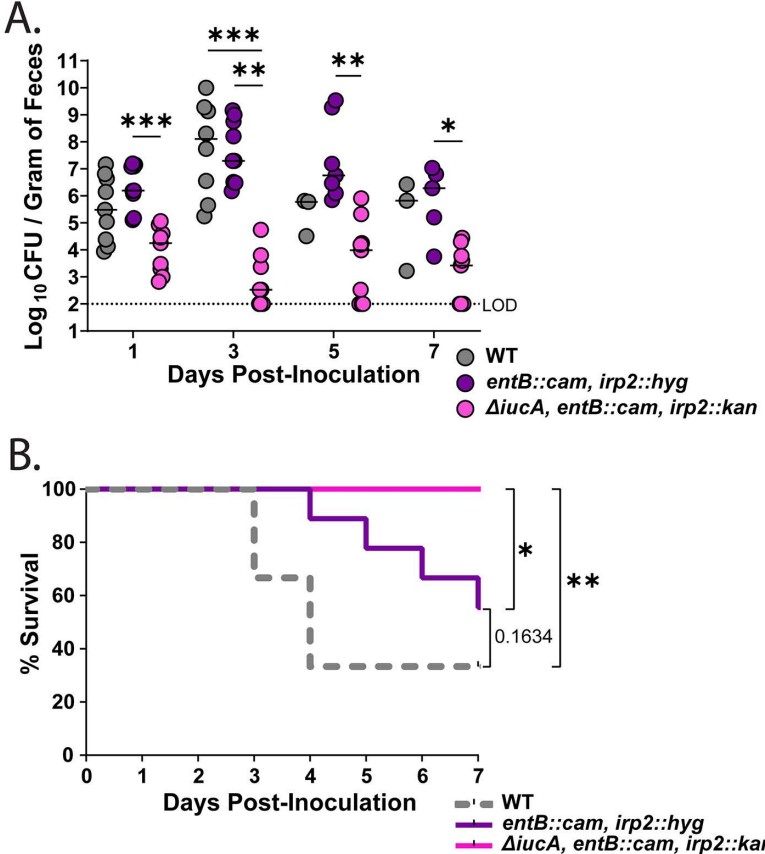

**Fig 3. Aerobactin is sufficient for gut colonization but other siderophores are required for increased virulence.** (A-B) C57BL6/J mice were orally inoculated with either the WT, an isogenic aerobactin-only synthesizing mutant (*entB::cam irp2::hyg*), or a siderophore-deficient mutant (*ΔiucA entB::cam irp2::kan*) and followed for up to 7 days. (A) Median fecal shedding for the WT and each of the isogenic mutants on the designated day, with each symbol representing a single mouse on the given day (n = 9 per group). The dotted line represents the LOD. A Kruskal-Wallis test, followed by Dunn's test of multiple comparisons was performed for statistical significance. (B) Kaplan-Meier curve indicating the percent survival of the mice. Log-rank (Mantel-Cox) tests were performed to compare survival odds. LOD, limit of detection. *$P < 0.05$, **$P < 0.01$, and ***$P < 0.001$; ns, not significant.

virulence compared to their respective wild-type parental counterparts (Fig 5C, 5D). Consistent with our observations with hvKP1 (S2B Fig), disrupting *iucA* in both hvKP2 and hvKP94 backgrounds significantly reduced total siderophore output (Fig 5E). These results reinforce that while aerobactin is dispensable for gut colonization, it is a conserved driver of translocation and virulence across genetically distinct hvKP lineages.

### Aerobactin promotes hvKP adherence, invasion, and translocation

Since our results precluded a major role for the M cells in hvKP translocation (Fig 2C, 2D), and previous studies have established that hvKP isolates interact with colonic epithelial cells [13,41], we sought to understand the impact of aerobactin on hvKP interactions with intestinal epithelial cells using a colonic Caco-2 cell culture model. We did not observe a difference in the ability to adhere or invade between the WT and the *iucA-* mutant when grown in iron-replete media (LB) (S3A, B Fig). In contrast, when both strains were grown in conditions where aerobactin provides a growth advantage (LB-DIP), we observed a significant reduction in both adhesion and subsequent invasion of the epithelial cells by the *iucA-* mutant in comparison to the WT strain (Fig 6A, 6B). When adjusted for adhesion, the defect in invasion was masked

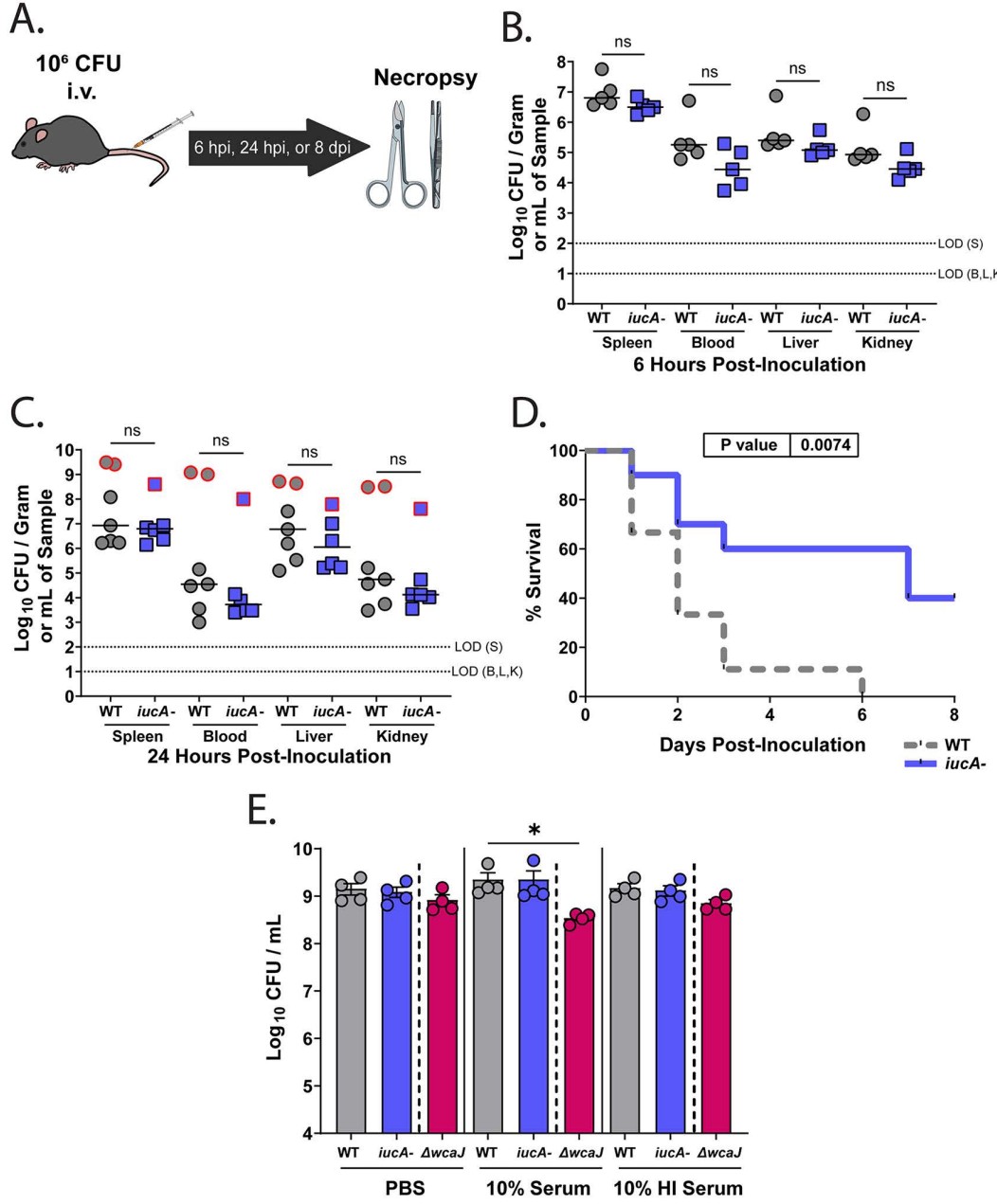

**Fig 4. Aerobactin is dispensable during early colonization events but contributes to virulence in a systemic model of infection.** Eight-week-old C57BL6/J mice were injected with $10^6$ CFU in 100 μL of either the WT or the isogenic *iucA-* mutant via the lateral tail vein. (A) Schematic of the murine systemic infection model. (B-C) Bacterial burden in the extra-intestinal sites (spleen, blood, liver, and kidneys) was either examined at 6 hours (B) or 24 hours (C) PI (n = 5-7 per group). Symbols with a red outline represent mice at an *in-extremis* state (severe illness) at 24h PI. Groups were compared using the Mann-Whitney *U* tests. (D) Survival analysis of mice infected via lateral tail vein with either WT (dashed gray line) or the *iucA-* mutant (solid blue line). Survival was monitored for 8 days, and statistical significance determined using a Log-rank (Mantel-Cox) test with P-values indicated. ns, not significant. (E) WT (gray), *iucA-* (blue) or KPPR1S Δ*wcaJ* (capsule-less control [magenta]) grown to mid-log phase ($10^7$ CFU) in LB + 200 μM DIP (iron-chelated media) were exposed to 10% rabbit serum, 10% heat inactivated (HI) rabbit serum, or PBS-control, for 1 hour at 37°C to determine survival. Shown is the mean ± SEM (n = 4) each assay was performed in triplicate and statistical significance was determined using the Kruskal-Wallis test, followed by Dunn's test of multiple comparisons. LOD, limit of detection; PI, post-inoculation. *$P < 0.05$, ns, not significant.

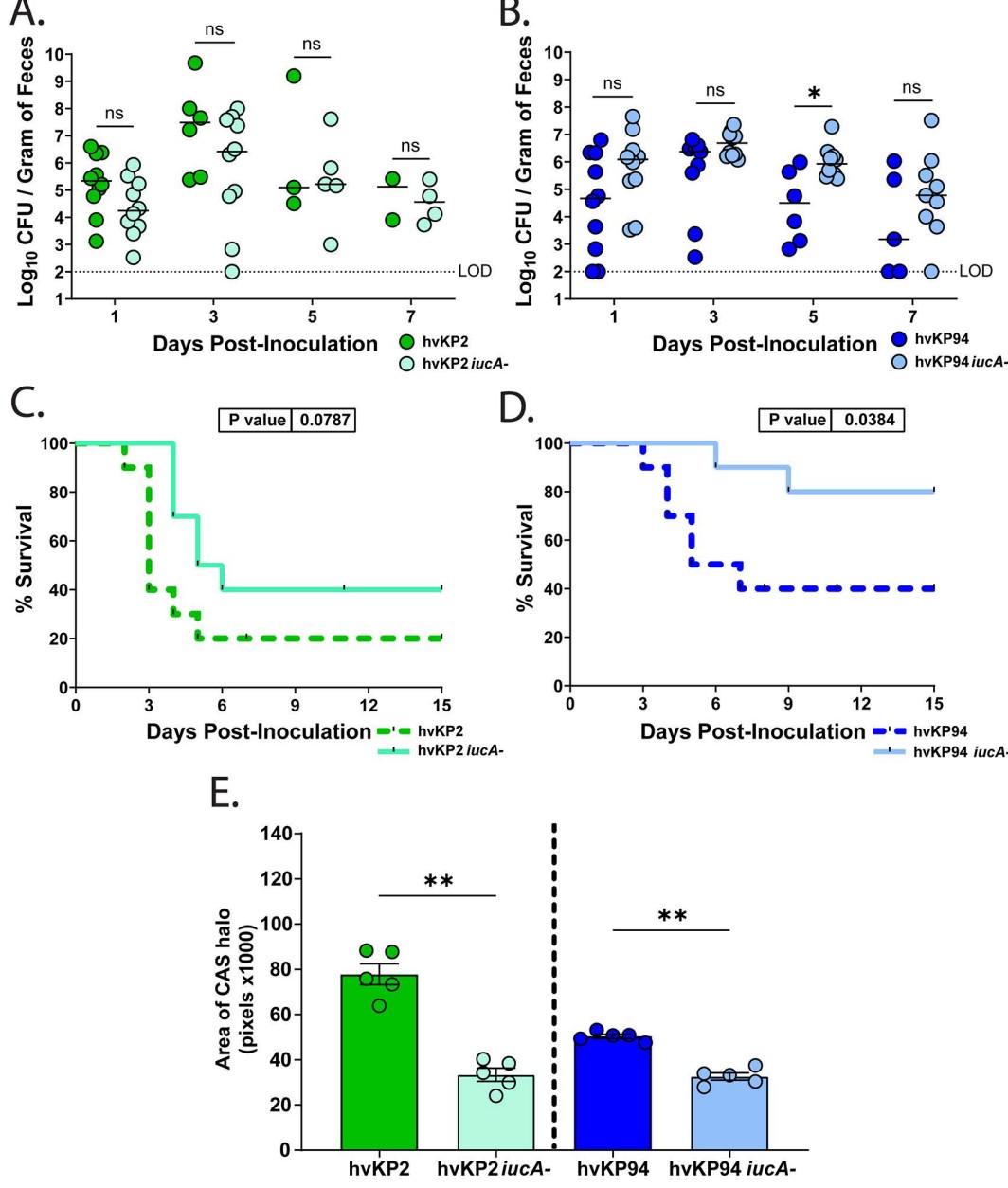

**Fig 5. Aerobactin impacts virulence without affecting hvKP gut levels across genetically distinct isolates.** (A-B) C57BL6/J mice were orally inoculated with either the hvKP2 (KL1 isolate) or the isogenic *iucA-* mutant (A) or hvKP94 (KL1 isolate) and its isogenic *iucA-* mutant (B), and fecal shedding was used to measure gastrointestinal bacterial burden. All strains were grown overnight in LB, and mice inoculated with ~$10^6$ CFU. Median shedding is shown for respective groups on the given day (n = 10 per group). A dotted line indicates the LOD. Statistical analysis was done using the Mann-Whitney *U* tests. (C-D) Survival analyses of mice colonized in the gastrointestinal tract with hvKP2 (C) or hvKP94 (D) (dashed lines) and their *iucA-* isogenic mutant counterparts (solid line) are shown. Survival of mice was monitored for 15 days PI and an *in-extremis* state or death was scored as non-survival. Statistical significance was determined using the Log-rank (Mantel-Cox) tests with *P*-values indicated. (E) Total siderophore production was assessed on CAS agar plates for either hvKP2 or hvKP94 and the respective *iucA-* isogenic mutants. Each strain was grown in LB overnight and diluted 1:10 in PBS before plating. Halos were measured and expressed as the mean area in pixels x1000. Shown are the averages for each experiment (n = 5). Statistical significance was determined using the Mann-Whitney *U* tests. LOD, limit of detection. *$P < 0.05$;, **$P < 0.01$, ns, not significant.

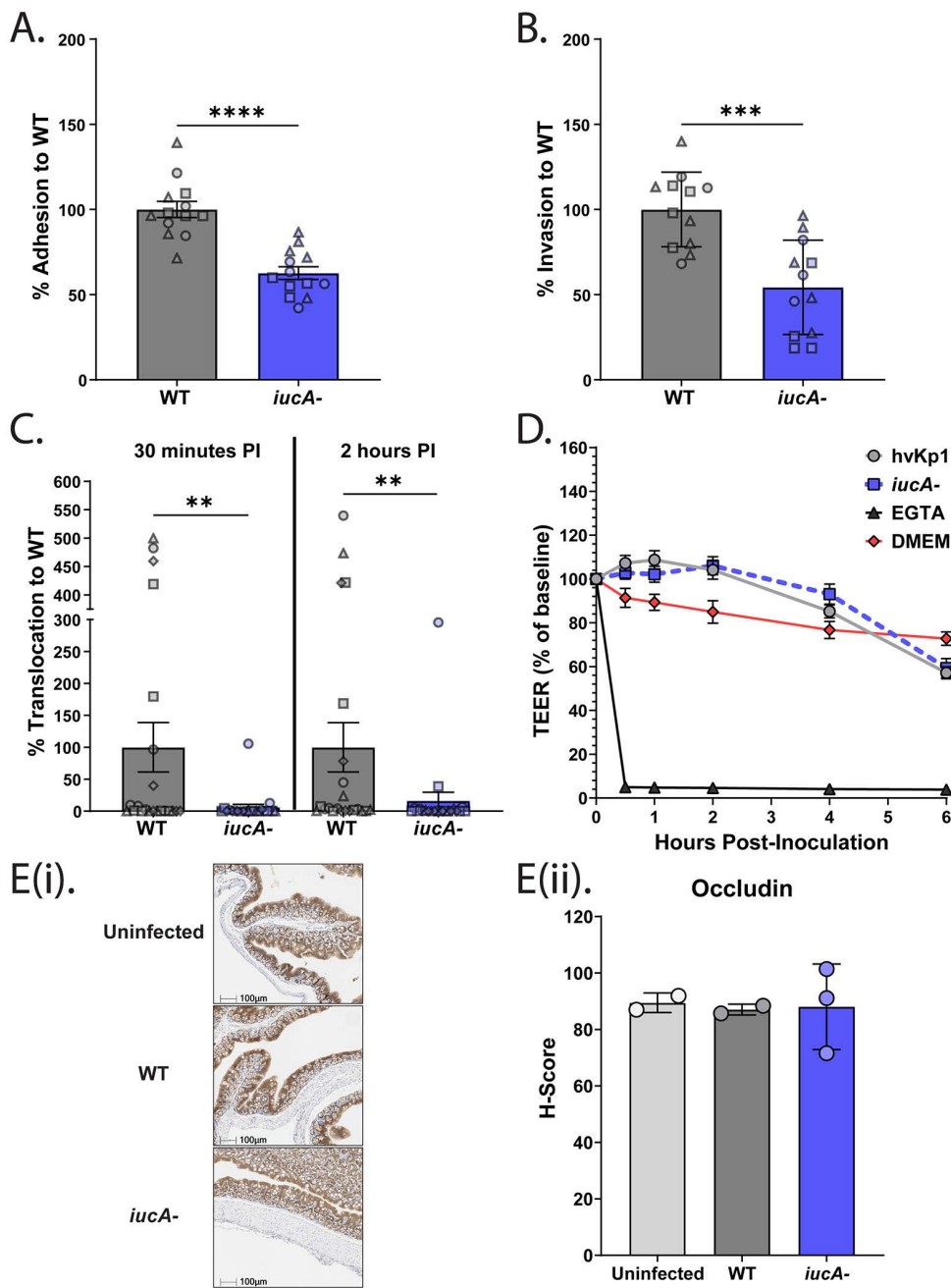

**Fig 6. Aerobactin promotes hvKP1 adherence, invasion, and translocation.** (A-B) Caco-2 (human intestinal epithelial cells) monolayers were infected with either the WT (gray) or the *iucA-* mutant (blue) grown to mid-log phase in LB+200 µM DIP (iron-chelated media) to test for adhesion (A) or invasion (B) at an MOI 50. Data are expressed as percent to WT, and the results of 3 independent assays, each with 3-5 replicates, are shown as mean±SEM. Statistical significance was determined using the Mann-Whitney *U* tests. (C-D) Polarized Caco-2 cells were grown on transwell inserts and allowed to differentiate for 19 days before infecting with either WT (gray) or *iucA-* (blue) grown to mid-log phase in LB+200 µM DIP. The top and bottom chambers were sampled at 30min and 2h PI to measure for translocation (C), with data expressed as percent to WT. The results of 4 independent assays, each with ≥ 5 replicates, is shown and Mann-Whitney *U* tests were performed for statistical significance. (D) TEER measurements were obtained using a Millicell ERS-2-Volt-ohmmeter at the indicated time points and plotted as mean percent of baseline (starting value). Wells treated with EGTA or DMEM served as experimental controls. The pooled data from 5 independent assays are shown±SEM, each with ≥4 replicates. (E). Representative immunohistochemistry (IHC) images (i) of murine proximal colons (5 µm sections) that were collected 24h PI from mice inoculated with WT or *iucA-* mutant or from uninfected vehicle-control mice (n≥2 per group) and stained for Occludin (1:400), and quantified H-scores (0-300) (ii) detailing Occludin expression via Halo image analysis. PI, post-inoculation. *P<0.05, **P<0.01, ***P<0.001, and ****P<0.0001.

(S3C Fig), indicating that the observed invasion defect is primarily due to a defect in adhesion, which is considered a critical first step in translocation. The same effect in adhesion and invasion was observed in the hvKP2 and hvKP94 backgrounds (S3 D, E Fig). Furthermore, this defect in adhesion which impacts invasion was rescued upon *trans*-complementation of the *iucA*- mutant with plasmid pFUS2[*iucA-D*] containing the *iuc* operon, which partially restored siderophore production [17] (S3 F-H Fig). Subsequently, we used Caco-2 cells grown on transwell inserts to assess whether the reduction in adhesion and invasion affected bacterial translocation through a monolayer, by measuring CFU that have crossed through the epithelial barrier. Compared to the WT, the *iucA*- mutant strain had a defect in translocation (Fig 6C), suggesting that the initial adhesion defect likely results in reduced translocation-ability of the *iucA*- strain. *Trans*-complementation with pFUS2[*iucA-D*] once again rescued the defect in translocation (S4A-D Fig).

Moreover, our transwell assay allowed us to provide insight into hvKP translocation route. Two well-established routes of translocation are known: transcellular (moving through cells) and paracellular (moving between cells). To assess which route was being utilized, we measured tight junction integrity using transepithelial electrical resistance (TEER). Interestingly, no differences in TEER values between the WT and *iucA*- mutant strains were identified (Fig 6D), suggesting that translocation likely initiates through the transcellular route. To confirm the effects of hvKP on tight junction proteins *in vivo*, we examined the expression of Occludin, a key protein in tight junction formation and function [42,43], in murine colonic tissue. The expression of Occludin in both WT-infected and *iucA*- mutant-infected mice was similar to the expression in naïve (control) mice (Fig 6E), indicating that the tight junction proteins were unperturbed by hvKP during early infection (24 hours PI), with or without aerobactin. Histological examination of the hematoxylin and eosin (H&E)-stained colonic tissues showed only minor changes in mice infected with the WT or the *iucA*- mutant compared to the naïve mice, as indicated by both image analysis and blinded scoring (S4E-Fii Fig). Taken together, our data illustrate the role of aerobactin primarily in adherence to epithelial cells, a prerequisite step for translocation through the transcellular route.

### Aerobactin deficiency affects the HMV phenotype in hvKP

Bacterial surface features play a crucial role in engagement with the host environment [44,45]. Fimbrial structures on the bacterial surface promote adhesion and, in turn, facilitate host cell invasion [46]. *Kpn* expresses two major types of fimbriae: type 1 and type 3. The type 1 fimbriae mediate adhesion to host epithelial cells in the respiratory and urinary tracts, whereas the type 3 fimbriae are linked to biofilm formation and adherence to abiotic surfaces under static conditions and both type 1 and 3 have been implicated in catheter associated bladder infection [47]. To assess whether aerobactin influences fimbrial expression under iron-depleted conditions, we measured the expression of *fimI* and *fimA*, which encode a putative anchor subunit and the major shaft protein of the type 1 fimbriae, respectively. We also analyzed the expression of *mrkA* and *mrkB*, which encode the major pilin subunit and a chaperone protein important for type 3 fimbrial assembly. Transcriptional analysis via qRT-PCR revealed that although all four genes were upregulated under low-iron conditions, deletion of *iucA* did not affect their expression (S5A Fig). These results suggest that the reduced epithelial cell engagement observed in the *iucA*- mutant is likely independent of fimbrial gene expression.

Another surface feature that can impact adherence to epithelial cells is capsular polysaccharide (CPS). Capsule has been shown to interfere with adherence in multiple pathogens [48–50]. Thus, we explored whether aerobactin contributes to CPS and the related hypermucoviscous (HMV) phenotype of hvKP. HMV, a hallmark of hvKP isolates, is characterized by increased mucoidy due to longer CPS chain-length distribution [10] and is regulated by the *rmpD* gene [51] within the *rmp* locus (Fig 7A). To test whether iron levels impact expression of the CPS and the *rmp* loci, we conducted qRT-PCR analysis using RNA isolated from the WT and the *iucA*- mutant grown in either iron-replete or iron-depleted media. Genes involved in capsule biosynthesis and retention (*galF*, *wzi, and manC*) and the *rmp* locus (*rmpA, rmpD,* and *rmpC*) were upregulated in iron-depleted conditions, with an even more pronounced increase observed in the *iucA*- mutant (Fig 7B, 7C). Sedimentation assays, revealed significantly increased HMV levels in *iucA*- mutants compared to their WT counterparts (Figs 7D, S5B), while CPS levels, measured by uronic acid quantification, were unaffected (S5C Fig). Furthermore,

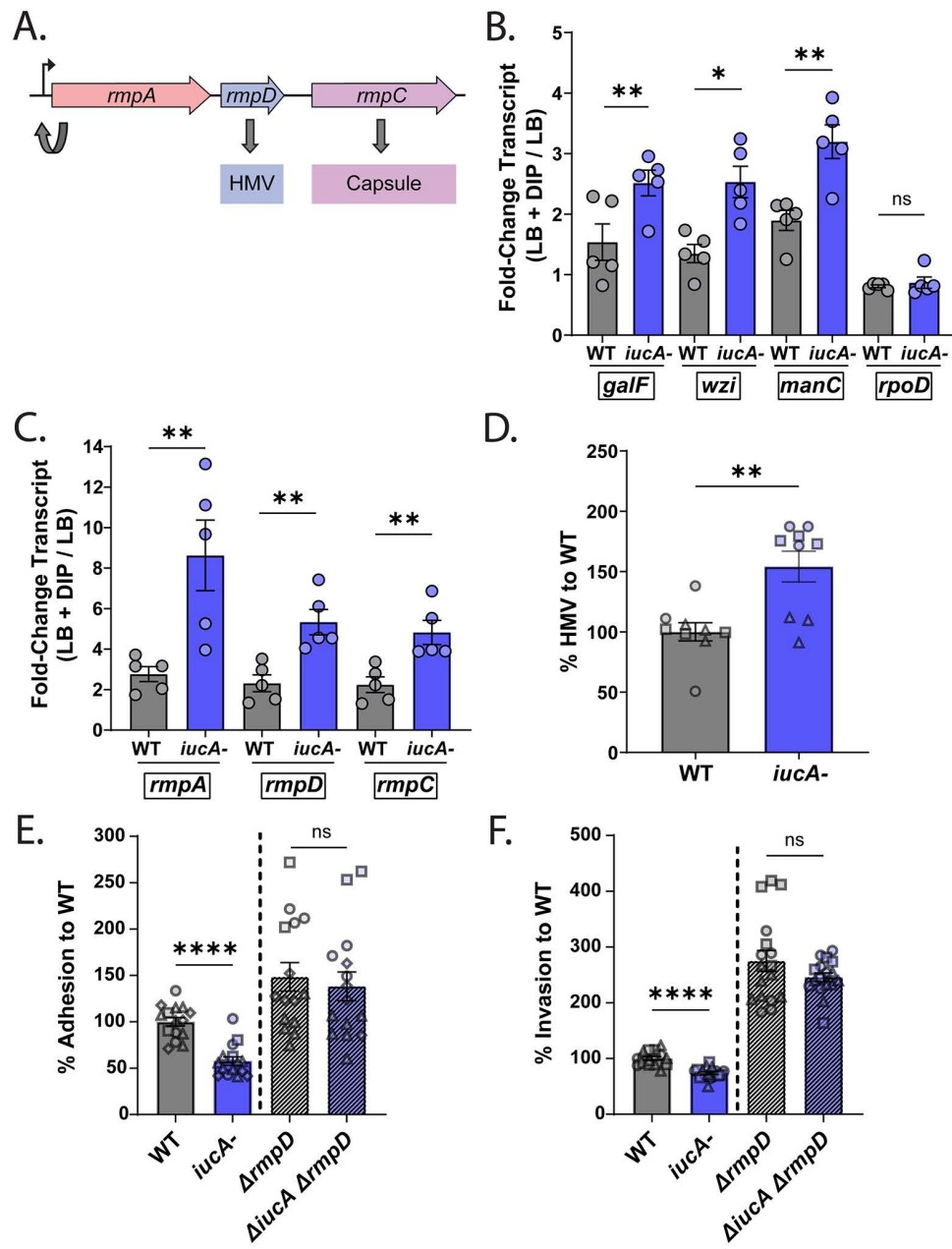

**Fig 7. Aerobactin expression under iron restrictive conditions regulates HMV levels.** (A) Schematic of the *rmp* operon of *K. pneumoniae*, with RmpA positively regulating the *rmp* locus, RmpD modulating HMV and RmpC positively regulating capsule biogenesis (B-C) qRT-PCR comparing gene expression of capsule locus genes (*galF*, *wzi, and manC*) with *rpoD* transcript levels shown as a control (B) and *rmp* locus genes (*rmpA*, *rmpD*, and *rmpC*) (C) in the WT and the isogenic *iucA-* mutant grown in LB+200 μM DIP (iron-chelated media) and LB (iron-rich media). Each biological replicate (n = 5) was run as a duplicate. The fold-change in transcripts is shown with *gyrA* as the internal control for $2^{-\Delta\Delta C_T}$. Statistical significance was determined using the Mann-Whitney *U* test. (D) HMV comparison via low-speed spin-down assay of the WT and *iucA-* mutant grown to mid-log phase (OD$_{600}$~0.5) in LB+200 μM DIP (iron-chelated media), expressed as the percent to WT. Shown are the results from 3 independent experiments (n = 3). (E-F) Caco-2 monolayers were infected with either the WT (gray), *iucA-* mutant (blue), *rmpD::kan* (dark gray), or the Δ*iucA, rmpD::kan* (dark blue) that were grown to mid-log phase in iron-chelated media and compared for adhesion (E) and invasion (F) at an MOI of 50, and expressed as a percent to WT±SEM. (D-F) The results from 3-4 independent assays are depicted (n ≥ 2). Mann-Whitney *U* tests were performed on the pooled samples of each experiment to determine statistical significance. *$P < 0.05$, **$P < 0.01$, ***$P < 0.001$, and ****$P < 0.0001$; ns, not significant.

*trans*-complementation of the *iuc* operon restored HMV levels to those observed in the WT strain (S5D Fig), implicating aerobactin in modulating hvKP HMV phenotype.

Based on our HMV results, we focused on RmpD, as it is known to directly interact with Wzc, a tyrosine kinase involved in capsule polymerization and export, leading to longer CPS chains characteristic of HMV [10,52]. Previous work showed that an *rmpD* mutant deficient in HMV exhibited increased adherence to J774A.1 macrophage cells [51]. Therefore, we hypothesized that if the reduced adherence observed in the *iucA-* mutant is due to increased HMV mediated by RmpD, then a double mutant lacking both *iucA* and *rmpD* should display similar adherence levels similar to an *rmpD-* mutant. Compared to the WT, we observed an increased level of adhesion and corresponding invasion in our cell culture model with an *rmpD::kan* (*rmpD-*) mutant (Fig 7E, 7F), consistent with previous data using J774A.1 macrophage cells [51]. Furthermore, the Δ*iucA rmpD-* mutant behaved similarly to the *rmpD-* mutant (Fig 7E, 7F), suggesting that dysregulation of the *rmp* locus in the absence of aerobactin contributes to reduced adherence and invasion of the mutant. Taken together, our gene expression and cell culture data demonstrate that aerobactin-mediated iron acquisition plays an important role in modulating capsular polysaccharide and HMV gene expression, thereby influencing hvKP interactions with epithelial cells.

## Discussion

Although the GI tract is a recognized reservoir for *Kpn* [6,29,53], with both pathotypes implicated in gut colonization in a hospital [6,7] and community setting [54], the mechanisms that enable translocation to sterile sites remain poorly understood. Using a murine model of *Kpn* gut colonization, our study is the first to provide molecular insight into the translocation of hvKP, identifying aerobactin-mediated metal homeostasis as critical for dissemination and systemic virulence.

Previous studies have highlighted the importance of siderophores in *Kpn* pathogenesis, particularly in nasopharyngeal and pulmonary infection, where evasion of the host innate immune protein lipocalin 2 (Lcn2) via yersiniabactin and salmochelin enhances virulence [22,55,56]. Furthermore, the siderophores produced by the *Kpn* isolate KPPR1 (enterobactin, salmochelin, and yersiniabactin) were shown to be important for inflammation and for dissemination from the lung to the spleen [22]. However, the lung and the gut are fundamentally distinct anatomical sites, and whether siderophores contribute to gut colonization and dissemination has not been the focus of previous studies. Our findings demonstrate that hvKP expresses multiple siderophores in the GI tract, and while aerobactin is not required for gut colonization (Fig 1D), overall siderophore production is critical. A siderophore-deficient strain showed poor colonization, whereas a strain expressing aerobactin alone was sufficient to colonize the GI tract (Fig 3A), indicating that siderophore-mediated metal acquisition is essential for robust colonization. Furthermore, the *iucA-* mutant showed 90% survival of inoculated mice and the aerobactin-only expressing strain showed ~50% survival, whereas the WT strain caused 70–80% mortality post-gut colonization. These data imply that while aerobactin is the major contributor to hvKP virulence, it requires all four siderophores for full virulence. These results on siderophore-mediated virulence are in line with other studies, which have shown that aerobactin under certain growth conditions accounts for ~90% of all siderophores produced by hvKP [17] and is the major contributor of siderophore-mediated virulence in a subcutaneous challenge, intraperitoneal model, and pneumonia model of infection [21,57,58]. Interestingly, our coinfection studies with the WT and the *iucA* mutant revealed that both strains generally colonize equally well in the GI tract, likely because the mutant has an intact aerobactin receptor (*iutA*), allowing it to uptake the aerobactin synthesized by the WT (trans complementation; social cheater). However, at extra-intestinal sites, we observed the WT at a higher density than the *iucA* mutant, suggesting that the WT is unable to mask the mutant's defect at the level of translocation, possibly because of host-associated bottlenecks at the translocation step.

Pathogenic bacteria employ strategies to overcome host barriers and translocate from the gut to sterile sites, causing invasive disease. For bacteria that translocate transcellularly, adherence to epithelial cells is considered the initial step in the translocation process. In *S. Typhimurium*, the fimbriae contribute to epithelial cell adherence, and the type III secretion system-1 (T3SS-1) and several other outer membrane proteins are critical for internalization [59]. However, *K. pneumoniae* strains do not encode a T3SS but nonetheless are able to adhere to and invade epithelial cells [13]. Bacterial factors,

such as the Sap (sensitivity to antimicrobial peptides) transporter [41] and the type VI secretion system (T6SS) [60], have been demonstrated to be critical for hvKP interactions with epithelial cells. However, these factors are also required for gut colonization and thus are not exclusive to the translocation process. We focused on aerobactin, a siderophore present in other *Enterobacteriaceae*, as it has been established as a known contributor to *Kpn* hypervirulence. In pathogenic *E. coli* isolates, the role of aerobactin in colonization and invasive disease is contradictory. In one study, aerobactin from a Shiga toxin-producing *E. coli* (O104:H4) was required for gut colonization in streptomycin-treated mice [61], whereas in germ-free lambs, aerobactin from *E. coli* isolate 31A (serotype O154:H⁻) was dispensable for gut colonization, but important for systemic spread [62]. Herein, we establish that hvKP translocates via the transcellular pathway without affecting gut permeability and aerobactin contributes to the translocation step without impacting gut colonization. This phenomenon was conserved across the different genetic backgrounds tested, where respective *iucA*- mutants in diverse genetic back-grounds colonized as well as their WT counterparts (Fig 5A-5B) but had reduced, not abolished, virulence (Fig 5C-5D). In concordance with Russo et al.'s results [63], the hvKP2 isogenic *iucA* mutant still had high virulence, possibly due to the expression of the genotoxin colibactin, which hvKP1 does not express or to the high expression of the other siderophores that likely offset for the loss of aerobactin. Collectively, these data demonstrate that, while aerobactin is a major contributor, other hvKP factors likely contribute to translocation and virulence.

*Kpn* is typically regarded as an extracellular pathogen. However, recent studies have shown that *Kpn* can internalize and replicate within epithelial cells and hepatic macrophages, suggesting an intracellular lifecycle phase [13,64]. *Kpn* utilizes the Rho family GTPases and phosphatidylinositol 3-kinase (PI3K)/Akt signaling to subvert the host cytoskeletal function and invade host cells [13]. Furthermore, our translocation results are consistent with previous data [13] that sug-gest that *Kpn* translocates through the transcellular pathway and does not impact the expression of tight junction proteins or the integrity of epithelial cell monolayers. However, in a *Kpn* lung infection model, siderophores were shown to stabilize host hypoxia inducible factor 1 α (Hif-1α) in alveolar epithelial cells, which controls genes involved in inflammation and vascular permeability, and subsequently promoted *Kpn* dissemination from the lung to the spleen [22]. Additionally, siderophore-mediated Hif-1α stabilization has also been observed with enteric pathogens, suggestive of a conserved molecular interaction at the host mucosal surface. Whether siderophore-mediated Hif1α stabilization occurs during hvKP colonization of the intestinal tract, needs further exploration.

Colonization by *Kpn* of the gastrointestinal tract has been thought to be primarily asymptomatic, without overt inflamma-tion [29]. In agreement with these findings, our data show that when colonized with hvKP, the gut displays minor disrup-tions, with only subtle epithelial damage and goblet cell depletion observed when colonized with the WT or *iucA*- mutant, respectively (S4Fi, ii Fig). In contrast to *Kpn*-associated lung infection, where siderophores trigger inflammation [22], hvKP gut colonization caused minimal histopathological changes, with no major differences observed between WT and *iucA*- strains. However, further molecular studies are needed to understand the immune response to hvKP gut colonization and whether siderophores influence the host inflammatory response.

Transition metals, such as iron, play a critical role for bacteria by serving as a co-factor for various enzymes involved in key biological functions. Both the host and bacteria tightly regulate iron levels, as high iron levels can be toxic due to the generation of free radicals [65]. Under iron-restrictive conditions, siderophore and capsule expression in *Kpn* increases via the regulatory effect of the ferric uptake regulator (Fur), which tightly controls the expression of dozens of genes that form the Fur regulon [66,67]. Additionally, the Apo form of IscR directly binds to the aerobactin locus to positively regulate aero-bactin expression under iron-restrictive conditions [68]. In many hvKP isolates, including hvKP1, aerobactin is the primary siderophore [17], and its expression through Fur and IscR likely allows hvKP to fine tune intracellular iron levels, which can have a profound effect *on Kpn* pathogenesis.

The *iucA*- mutant had a defect in engagement with epithelial cells under low-iron conditions, prompting an examination of hvKP surface features. While type 1 and 3 fimbriae are known contributors of *Kpn* interactions with host cells [69,70] their expression were similarly upregulated in both the WT and *iucA*- mutant, the latter exhibited higher gene expression

of CPS synthesis and *rmp* loci, likely due to increased iron stress and precluding the fimbriae from the observed differences in adhesion. We observed a corresponding increase in HMV levels in the *iucA-* mutant but not in CPS quantity, suggesting that high gene expression does not necessarily lead to a phenotypic change to capsule levels. The small protein RmpD is known to regulate capsule chain length [10], and *rmpD* deletion only affects HMV levels [51]. Notably, the *rmpD-* mutant had greater binding affinity to epithelial cells, and the Δ*iucA rmpD-* double-mutant compensated for the *iucA-* defect (Fig 7E, 7F). These results highlight the importance of aerobactin-mediated iron homeostasis, which is essential for the regulated expression of the *rmp* locus. In turn, controlled expression of HMV through the *rmp* locus is crucial for hvKP interactions with host cells, influencing its virulence potential. There is precedence for this line of thinking as *Kpn* is known to modulate CPS expression while residing in *Klebsiella*-containing vacuoles within macrophages, enabling adaptation to the changing host environment [71]. *Kpn* also responds to arginine present at the host mucosal sites, and through ArgR, positively regulates the expression of the *rmp* locus, resulting in an increase in mucoidy, which allows *Kpn* to evade macrophages [72]. Additionally, growth in urine was associated with a reduction in the HMV phenotype, which was attributed to Wzc, a tyrosine kinase that regulates CPS polymerization and extrusion [73]. Together, these studies and our data suggest that hvKP carefully modulates HMV expression in response to the host environment to promote virulence.

One of the caveats for our study was the inability to successfully complement *iucA* at the native chromosomal site. However, we were able to replicate our CAS agar, HMV phenotype, adhesion, invasion and translocation studies through trans complementation with the pFUS2[iucA-D] plasmid, although the plasmid is not retained by hvKP1 in the murine gut making it unfeasible for *in vivo* studies. Additionally, the *iucA-* strain has been extensively used and sequenced to rule out other mutations besides the *iucA* deletion, suggesting that our observed *in vivo* and *in vitro* phenotypes are aerobactin dependent. Another caveat of our study is that we tested 3 genetically distinct hvKP isolates (hvKP1, 2 and 94), all encoding aerobactin, which has been observed to be present in 98% of all hvKP isolates [74]. However, *Kp* isolates without aerobactin are also known to cause severe disease, and convergent isolates that contain aerobactin compared to hvKP isolates have been observed to have reduced virulence in murine model of infections, suggesting that factors besides aerobactin such as CPS, HMV and other encoded factors present on the virulence plasmid contribute to hvKP associated disease states [75–77]. Additionally, while our cell culture assays provide molecular insights into the hvKP translocation process, *in vivo* immunofluorescent studies are required to further elucidate adhesion, invasion, and translocation.

In conclusion, while aerobactin has long been recognized as critical for hvKP-mediated invasive disease [17,21,63,78], our study identifies its previously uncharacterized role in facilitating translocation from the gut. Pathogenic bacterial translocation, a key step in the progression toward invasive disease, is a complex process involving both host and bacterial factors, and this study advances our understanding of the process. Moreover, our findings shed light on the impact of aerobactin on the virulence of hvKP strains from diverse genetic backgrounds and link aerobactin-mediated virulence to intestinal barrier function, making aerobactin an attractive molecular diagnostic marker for assessing the risk of development of invasive disease in a clinical setting. Targeting hvKP strains has become increasingly urgent with the emergence of convergent hypervirulent and multi-drug-resistant (MDR) strains [79–82]. Future research targeting translocation determinants, including siderophore-mediated processes, could offer novel approaches to combating the spread of hvKP infections.

## Materials and methods

### Ethics statement

This study was conducted in accordance with the guidelines outlined by the National Science Foundation's animal welfare requirements and the Public Health Service Policy on Humane Care and Use of Laboratory Animals [83]. All murine studies were conducted according to the American Association for Laboratory Animal Science (AALAS) guidelines and were approved by the Wake Forest Baptist Medical Center Institutional Animal Care and Use Committee (IACUC). The approved protocol numbers for this project are A20-084 and A23-064.

## Bacterial strains

Strains, plasmids, and primers used in this study are listed in S1–S3 Tables, respectively. The in-frame deletion of *iucA* in hvKP1 (WT) (ST86, KL2; bacteremia, hepatic and splenic abscess; Buffalo, New York, USA) background was generated as described previously [84]. Briefly, the kanamycin (kan) cassette with flanking FLP recombinase target sites (FRT) was amplified from pKD4 with 50 bp homology to *iucA* at the 5' and 3' ends, using Q5 polymerase (M0491L; New England Biolabs), and treated with DpnI (1 µL/ 100 µL PCR product) to digest the pKD4 template. The purified PCR product was electroporated (1.8 kV, 400 Ω, 25 µF) into hvKP1 containing the temperature-sensitive pKD46 plasmid, which encodes the λ red recombinase genes under the control of an arabinose-inducible promoter (pBAD). Successful mutants were selected on lysogeny broth—Lennox (LB) agar plates supplemented with kan (25 µg/mL). The presence of the kan cassette was confirmed via PCR. The kan cassette was removed with Flp recombinase using pFlp3 [85] to generate the Δ*iucA* strain. The plasmid complemented strain was generated by transforming hvKP1 *iucA::kan* with the pFUS2[*iucA-D*] plasmid [17]. Empty pFUS2 vectors were transformed into both WT and hvKP1 *iucA::kan* for direct comparison.

To construct the siderophore-deficient strain, Δ*iucA entB::cam irp2::kan*, in the hvKP1 background, an *irp2::kan* mutant was first generated in the Δ*iucA* background, following the same λ red recombinase process listed above, but with a 60 bp homology for the *irp2* gene at the 5' and 3' ends. Subsequently, the *entB::cam* cassette was amplified using Q5 polymerase (M0491L; NEB) from isolated genomic DNA from an *entB::cam* mutant in MKP103 [86]. The purified PCR product was electroporated into Δ*iucA irp2::kan* background strain containing pKD46. Successful mutants were selected on LB agar supplemented with chloramphenicol (cam) (50 µg/mL), confirmed via PCR, and plasmid cured at 37°C.

Generation of the *rmpD* mutants in the WT and Δ*iucA* backgrounds involved amplification of the *rmpD::kan* cassette with flanking FRT sites from pKD4 with 60 bp homology to *rmpD* at the 5' and 3' ends, using Q5 polymerase (M0491L; NEB), and treated with DpnI (1 µL/ 100 µL PCR product) to digest the pKD4 template. As described above, the purified PCR product was used with WT and Δ*iucA* strains containing pKD46 for λ red recombination. The resulting mutants were selected on LB agar supplemented with kan (25 µg/mL) and confirmed via PCR.

The hvKP2 and hvKP94 *iucA::kan* mutants (AZ235 and AZ255) were generated using λ red recombinase via pKD46 to disrupt the *iucA* gene with the kan cassette from pKD4, as described above. Recombinants were selected on LB agar supplemented with kan (25ug/mL) and confirmed via PCR. The sequence type (ST) and capsule type (KL) for the parental hvKP2 (ST23, KL1) and hvKP94 (ST23, KL1) isolates were derived from the original genomic analysis conducted by the source laboratory (Russo Lab, Buffalo, New York, USA).

The apramycin-resistant hvKP1, hvKP1 *attTn7::apra* (AZ314), was constructed by amplifying the apramycin resistance cassette from the *attTn7* site of AZ94 [87] and introducing it into hvKP1 (AZ121) via λ red recombination, as described above. Recombinants were selected on LB agar containing apramycin (apra) (50 µg/mL) and confirmed by PCR.

Unless otherwise specified, all strains were grown overnight in LB at 37°C with constant agitation. Overnight cultures were diluted to $10^7$ CFU/mL for mouse infections or subcultured 1:100 for other assays.

## Kinetic growth assays

Strains to be tested were grown overnight at 37°C with constant agitation in LB-Lennox, $OD_{600}$ adjusted to 3, and then subcultured 1:100 in either LB or LB with 200 µM 2,2'-dipyridyl (DIP) to chelate iron. Subcultures were grown at 37°C with constant agitation, and at the designated time points, a sample was removed, diluted, and plated for enumeration.

For *in vitro* competitive growth studies, overnight cultures were prepared as above. Competing strains with distinct antibiotic selection markers were diluted 1:100 into fresh test media, and grown at 37°C with constant agitation. Samples were collected at designated time points, and plated on selective antibiotic plates for enumeration of each strain. The

Competitive Index (CI) was calculated for each time point using the following formula, where t(0) is the CFU/mL at the time of initial subculture and t(x) is the CFU/mL at different times post-initial subculture.

$$\text{Log}_{10}\ \text{CI}\ =\ (\text{Mutant } t(x)/\ \text{WT } t(x)/\ (\text{Mutant } t(0)/\ \text{WT } t(0))$$

## Mucoviscosity assay

Mucoviscosity levels were measured using the low-speed spin-down assay, previously described [88], with slight modifications. Strains to be tested were grown overnight in LB as described above, subcultured 1:100 in fresh LB with or without 200 μM DIP, and further grown at 37°C with constant agitation until mid-log ($OD_{600} \sim 0.5$). Samples were then adjusted to $OD_{600}$ of 0.5 in a final volume of 1.2 mL, and $OD_{600}$ measurements were recorded in triplicate. Afterward, 1 mL of each sample was centrifuged (1,000 x g for 5 min), and $OD_{600}$ measurements were recorded again in triplicate. The ratio of the average of the three measurements post-centrifugation over the average of the three measurements before centrifugation was quantified. Strains transformed with pFUS2 plasmids were grown in media supplemented with gentamicin to prevent plasmid loss.

## Uronic acid quantification assay

Uronic acid (UA) content, as an indicator of capsular polysaccharide amount, was measured as previously described [88,89] with slight modifications. Overnight cultures were subcultured 1:100 in the indicated media and grown to mid-log at 37°C with constant agitation. UA was extracted from each 500 μL sample by adding 100 μL of 1% Zwittergent 3–08 in 1M citric acid and incubating for 20 min at 50°C. Afterward, samples were centrifuged (21,000 x g for 5 min) at room temperature, and 300 μL of supernatant was incubated overnight with 1.2 mL of 100% ethanol (200 proof). Precipitates were centrifuged (21,000 x g for 10 min) at 4°C, the supernatants were removed, pellets were air-dried and subsequently resuspended in 200 μL of $H_2O$, and incubated for 10 min at 100°C with 1.2 mL of 12.5mM sodium tetraborate in sulfuric acid. Samples were then chilled on ice for 10 min and incubated with 20 μL 0.15% 3-phenylphenol. Each sample was loaded as triplicate on a 96-well plate, and absorbance was measured at 520 nm on a plate reader (Biotek H1 Synergy). UA levels were quantified against a standard curve with glucuronic acid.

## Mouse infections and sample collection

C57BL/6J specific pathogen-free (SPF) mice were obtained from Jackson Laboratory (Bar Harbor, ME) and bred and maintained at Biotech Place, Wake Forest Baptist Medical Center animal facility. For all studies, mice were randomly allocated to different groups, and the sample size was not predetermined.

Oral inoculations were performed as previously described [29]. Briefly, 5–7 week-old mice were fasted for 3–4 hours and orally fed 100 μL of *Kpn* resuspended to $\sim 10^6$ CFU/100 μL in 2% sucrose-PBS via pipette in two doses of 50 μL an hour apart. The inoculum dose was serially diluted and plated on selective media to enumerate and confirm the dose. For survival studies, animals were monitored for 7 days, and a state of in extremis or death resulted in the study endpoint.

For systemic infections, 8-week-old C57BL/6J SPF mice were injected intravenously with $\sim 10^6$ CFU of *Kpn* in PBS by tail vein inoculation. Following inoculations, animal health and body weight were monitored throughout the study period. Humane endpoints were defined by clinical signs of moribundity, including altered gait, hunched posture, increased respiration, or a failure to explore the cage when disturbed. While weight loss was monitored as an indicator of decline, it was assessed in the context of the animal's overall clinical stability; mice that remained active and responsive despite transient weight loss were closely monitored. Mice requiring euthanasia were marked as a death in the survival curve. All mice that survived up to day 8 post-infection were euthanized using $CO_2$ (2 L/min for 5 min) followed by cardiac puncture.

Bacterial shedding in the feces was quantified as previously detailed [29,30,89–91]. At designated time points, the bacterial burden in the gastrointestinal tract (distal ileum, cecum, and proximal colon) and extra-intestinal organs (mesenteric lymph nodes (MLN), liver, spleen, and kidneys) was determined as described. Briefly, samples were collected from mice following euthanasia, weighed, and diluted with 1X PBS. For the liver and kidneys, equal weight-to-volume of PBS was added, and all other organs were diluted 1:10. Diluted samples were homogenized using the Fisherbrand Bead Mill 24 Homogenizer at a strength of 3.1 for 2 min. Blood was collected from the submandibular vein using a 5 mm lancet. Samples were serially diluted and plated on antibiotic plates to enumerate CFU counts.

*In vivo* competition studies were conducted as previously described [29,30,89–91], with a 1:1 ratio of each strain resuspended in $10^6$ CFU/100 µL in 2% sucrose and orally fed to the mice as described above. Collected and homogenized samples were serially diluted and plated on selective antibiotic plates to distinguish the two strains. The CI was calculated using the following formula, where the output is the CFU/mL quantified in the sample and the input is the CFU/mL of the infection dose.

$$Log_{10}CI = (Mutant\ output/\ WT\ output)/\ (Mutant\ input/\ WT\ input)$$

## RNA extraction, cDNA synthesis, and RT-qPCR

RNA was isolated from *Kpn* strains from *in vitro* samples using the TRIzol method [92] with modifications [89]. Briefly, overnight cultures grown in LB were subcultured 1:100 into their respective media and grown to $OD_{600}$ ~0.5 before RNA isolation.

Bacterial RNA was isolated from murine cecal samples as previously described [30,89,91,93]. Briefly, cecal contents from mice colonized with the hvKP1 isolate (n = 4) were collected 4–6 days post-infection and placed in a 2 mL screw cap tube containing glass beads, diluted 1:1 with RNAlater (Invitrogen, AM7020), homogenized, and stored overnight at 4°C. An equal volume of chilled 1X PBS was added to the samples, and they were centrifuged (700 x *g* for 1 min at 4°C), and the supernatants were collected and centrifuged (900 x *g* for 5 min at 4°C). The pellets were resuspended in 500 µL 2X Buffer A (200 mM NaCl, 200 mM Tris base, 200 mM EDTA), 210 µL 20% SDS, and 500 µL UltraPure Phenol:Chloroform:Isoamyl Alcohol (25:24:1). Isolated crude RNA from both *in vitro* and *in vivo* samples were treated with DNase (Invitrogen; AM1907) followed by cDNA synthesis (iScript, BIO-RAD) [89,91,92]. Samples were purified via the Qiagen MinElute PCR Purification Kit.

RT-qPCR was performed as previously detailed [94,95] using *Kpn*-specific primers [89]. DNA gyrase (*in vitro*) and 16S rRNA (*in vivo*) served as reference genes with primers specific to *Kpn*. Samples were run in duplicate on a CFX384 Touch real-time PCR detection system (BIO-RAD). RNA expression was quantified using the $\Delta\Delta C_T$ threshold cycle (CT) method with fold-change calculated using $2^{-\Delta\Delta C_T}$ [89,94]. For *in vitro* samples, 3–5 biological replicates were tested in duplicate. For each *in vivo* sample, the average of two independent experiments was plotted.

## Adhesion and invasion assays

Adhesion and invasion assays were carried out as previously described [13], with slight modifications. Approximately 5 x $10^5$ Caco-2 [Caco2] (ATCC HTB-37) colonic epithelial cells in Dulbecco's Modified Eagle Medium (DMEM) supplemented with 10% heat-inactivated Cytiva HyClone fetal bovine serum (FBS), 1% Cytiva HyClone Non Essential Amino Acids (NEAA) 100X solution, 1% GlutaMAX supplement, and without antibiotics, were seeded in 1 mL volume per well in 24-well plates (Corning Costar Flat Bottom Cell Culture Microplates), and allowed to adhere and form monolayers overnight at 37°C with 5% $CO_2$. Cells were washed with 1 mL of 1X Dulbecco's Phosphate Buffered Saline (DPBS), and 1 mL fresh media without antibiotics was added to each well. *Kpn* was grown to mid-log in either LB or LB supplemented with 200 µM DIP, $OD_{600}$ adjusted to 0.5, and resuspended in FBS-free DMEM. Strains transformed with pFUS2 plasmids were grown in

media supplemented with gentamicin. Bacteria were added to each well in a multiplicity of infection (MOI) of 50 bacteria/cell in 250 µL. Plates were centrifuged at 200 x $g$ for 5 min. Samples of each inoculum were serially diluted and plated on antibiotic plates for CFU counts.

For adhesion assays, the centrifuged plates were incubated for 20 min at 37°C with 5% $CO_2$. Afterwards, the wells were washed three times with 1X DPBS, and the bacteria were recovered with 500 µL of 0.2% Triton X-100 (Sigma, X100-100ML). For invasion assays, plates were incubated as described above for 2 hours and subsequently washed twice with 1X DPBS. Cells were then treated with 1 mL of media containing 100 µg/mL gentamicin and incubated for another 2 hours to kill any extracellular bacteria. Cells were washed twice afterwards with 1X DPBS and intracellular bacteria were recovered with 500 µL of 0.2% Triton X-100.

Recovered bacteria were serially diluted and enumerated on selective media plates. The number of adhered or invaded bacteria was calculated as the proportion of the inoculum. Three to four independent experiments were conducted, each with ≥ 2 replicates. The data was expressed as a percent to WT.

## Translocation assays and TEER measurements

Translocation assays were performed as previously described [13,96–98] with modifications. Caco-2 cells (~$4\times10^5$ in 0.5 mL) in complete media (DMEM supplemented with 10% heat-inactivated FBS, 1% Gibco Antibiotic-Antimycotic (100X), 1% NEAA, 1% GlutaMAX supplement, and 50 µM 3-mercapto-1,2-propanediol) were seeded onto transwell inserts (3 µm pore size, 1.12 cm$^2$ surface area) in 12-well plates (Corning Transwell Multiple Well Plate with Permeable Polycarbonate Membrane Inserts) and allowed to differentiate for 19 days. *Kpn* strains were grown and resuspended in FBS-free DMEM at half the original volume to concentrate each inoculum. Cells were washed twice with media without antibiotics. Bottom chambers were subsequently filled with 1.5 mL of media without antibiotics, and top chambers (apical side of differentiated cells) were inoculated with 500 µL of bacteria (1 x $10^8$ CFU). Plates were incubated for 30 min as described above. Samples were collected from the top and bottom chambers, serially diluted, and plated on antibiotic plates to enumerate bacteria. The percentage of translocated bacteria was calculated as the proportion of the inoculum in the bottom chamber. Three independent experiments, each with ≥ 5 replicates, were conducted.

Strains transformed with either pFUS2 (empty vector) or pFUS2[iucA-D] (plasmid complement) were grown in LB supplemented with 10 µg/mL gentamicin overnight, and subcultured 1:100 into LB supplemented with 200 µM DIP and 5 µg/mL gentamicin, and grown to midlog. Afterwards, the strains $OD_{600}$ were adjusted to 0.5, and resuspended in FBS-free DMEM supplemented with 5 µg/mL gentamicin. Cells were also washed and replaced with media supplemented with 5 µg/mL gentamicin, prior to the start of the assay.

To examine tight junctions (TJs), Transepithelial Electrical Resistance (TEER) measurements were collected using a Millicell ERS-2-Volt-ohmmeter. TEER value >300 Ω/cm2 for the Caco-2 polarized monolayer was indicative of being fully differentiated. Only monolayers with baseline TEER measurements > 800 Ω/cm2 were utilized for these experiments. Assays were conducted as described above, and TEER measurements were conducted prior to infecting (T0) and at various time points post-infection (30 min, 1-, 2-, 4- and 6-hours). Ethylene glycol-bis(β-aminoethyl ether)-N,N,N′,N′-tetraacetic acid (EGTA; 10 mM), known to disrupt TJs [99], and DMEM-only were used as controls. Five independent experiments, each with ≥ 4 replicates, were conducted.

## Chrome azurol S (CAS) agar plate assay

The Chrome azurol S (CAS) agar plates were prepared as previously described [100,101] to detect siderophore production. Briefly, 100 mL of CAS dye was prepared by dissolving 10 mM $FeCl_3•6H_2O$ in 10 mL of 10 mM HCl and combined with 50 mL of 2 mM CAS dye in $H_2O$. The 60 mL mixture was subsequently combined with 40 mL of 5 mM Hexadecyltrimethylammonium bromide (HD™A). Separately, 500 mL of King Agar B (Sigma-Aldrich) was prepared. Both reagents were autoclaved for 15 min at 121°C. The cooled CAS dye was slowly added, mixed with the molten King Agar B, and poured into petri dishes.

To qualitatively detect siderophore production, overnight cultures of *Kpn* strains to be tested were grown at 37°C with agitation under aerobic conditions, diluted 1:10 with 1X PBS and 10 µL spotted on CAS agar plates. Plates were incubated for 16 hours at 37°C. An orange halo surrounding bacteria indicated a positive CAS reaction and the presence of siderophores. Halos were measured using ImageJ version 1.52a software by measuring the area of each halo while excluding the bacterial colony (NIH, USA, https://imagej.net/ij/) [102].

For hvKP94 parental and mutant strain, CAS agar plates were incubated for an additional 72 hours at room temperature to ensure accuracy of siderophore production potential. For assays conducted using the pFUS2 (empty vector) or pFUS2[iucA-D] transformed strains, CAS agar plates were incubated for an additional 24 hours at room temperature.

## Histology and imaging

C57BL/6J 5–7 week old mice were orally inoculated with 2% Sucrose PBS (vehicle-only control; n = 2), hvKP1 (WT; n = 2), or hvKP1 *iucA*- isogenic mutant (n = 3). All mice were euthanized 24 hours PI, and the proximal colons (approximately 4 cm) were collected and immediately flushed with 4% paraformaldehyde (PFA). Fecal pellets were carefully removed, and colons were fixed in 4% PFA for 72 hours. *Kpn* colonization density in the ileum and cecum at 24 hours was determined by plating tissue homogenates on selective antibiotic plates and enumerating CFU counts.

PFA-fixed proximal colon cross-sections were treated with 70% ethanol and embedded in paraffin and sectioned at 5 µm onto glass slides. Slides were dewaxed and stained with hematoxylin and eosin (H&E), using a routine H&E staining protocol. Briefly, H&E stains were performed using the autostainer XL from Leica Biosystems. The slides were stained with Hematoxylin (Epredia, 7211) for 2 mins and Eosin -Y (Epredia, 7111) for 1 min. Clarifier 2 (7402) and Bluing (7111) solutions from Epredia were used to differentiate the reaction. After staining, slides were then dehydrated in graded ethanol, ending in xylene, and coverslipped with Cytoseal 60 (Epredia, 83104). H&E stained slides were digitally imaged in the Aperio AT2 (Leica) using 20x objective. Blinded histopathological scoring of H&E slides was conducted for each sample.

To detect Occludin antigen, chromogenic Immunohistochemistry (IHC) was performed using the Leica Bond RX Autostainer system on paraffin-embedded mouse colon tissues that were sectioned at 5 µm onto positively charged slides. Slides were blocked and then incubated with Occludin Antibody (ab216327, Abcam) at 1:400 for 1h followed by ready to use Novolink Polymer (RE7260-CE, Leica Biosystems) secondary. Antibody detection with 3,3'-diaminobenzidine (DAB) was performed using the Bond Intense R detection system (DS9263). IHC stained slides were digitally imaged in the Aperio AT2 (Leica Microsystems) using 20x objective.

Occludin image analysis was performed using the HALO AI software (Indica Labs, version 4.0). The mucosa (epithelial cells) was separated from the underlying muscle layers using a MiniNet HALO AI tissue classifier. Next, a customized multiplex IHC algorithm was used to detect and quantify the number of cells positive for Occludin. Nuclei segmentation was performed using the AI default pre-trained algorithm with a hematoxylin weight of 1, occludin nuclear detection weight of 0.04, nuclear contrast threshold of 0.053, nuclear range of 11.3 to 144.81 µm, and nuclear segmentation aggressiveness of 0.55. An AI custom membrane segmentation algorithm detected cellular membranes with a maximum cytoplasm radius of 1.98 µm and membrane segmentation aggressiveness of 0.007. Occludin thresholds were set to 0.15, 0.4268, and 1.1161 for low, medium, and strongly staining cells, respectively. At least 42.2% of the membrane needed to be positive for the cell to be considered positive, since Occludin is mainly located at cell-cell junctions. Both the non-mucosal and mucosal tissue regions were analyzed.

## Serum survival/killing assay

To assess complement-mediated killing, *Kpn* strains grown to mid-log in LB supplemented with 200 µM DIP and adjusted to $OD_{600}$ 0.5, were washed, pelleted and resuspended in PBS to a concentration of $10^9$ CFU/mL. In a 96-well plate, 5 µL of each strain ($10^7$ CFU) were combined with 45 µL DMEM, and 5 µL of either rabbit serum (Cedarlane, #CL3111),

heat-inactivated (56°C for 30 min) rabbit serum (control), or 1x PBS (control). Following a 1-hour incubation at 37°C, bacterial survival was determined via quantitative enumeration of CFU.

## Statistical analysis

All statistical analyses were performed using GraphPad Prism 10.2.3 (GraphPad Software, Inc., San Diego, CA). Comparisons between two groups were analyzed using the Mann-Whitney $U$ test. For comparisons among multiple groups, the Kruskal-Wallis test was applied, followed by post hoc Dunn's multiple comparisons test or a one-way ANOVA, followed by Tukey's post-hoc test was performed. Competition experiments were analyzed using the Wilcoxon signed-rank test for matched pairs. Kaplan-Meier curves for survival comparisons were analyzed with the Log-rank (Mantel-Cox) test. The association between bacterial load in the cecal and extra-intestinal samples was assessed using simple linear regressions and a correlation matrix with Pearson correlation coefficients (r) at a 95% confidence interval. Associations between categorical variables in the 3x2 contingency table were assessed using the Fisher's exact test (Freeman-Halton extension). Data used to generate the figures is presented in S4 Table.

## Supporting information

**S1 Fig. Aerobactin provides hvKP a growth advantage *in vitro* under iron-limited growth conditions.** (A) Growth kinetics of the WT and *iucA-* mutant grown in LB (iron-replete media) or LB + 200 μM DIP (iron-chelated media) at 37°C under aerobic conditions, shown as (i) $\log_{10}$ CFU/mL over time. CFU counts for the WT and *iucA-* mutant in each growth condition were compared at 8 hours (i) and 24 hours (ii) PI using Mann-Whitney $U$ tests. (B) An *in vitro* competition assay between the WT (gray circles) and the *iucA-* (blue open circles) inoculated 1:1 in (i) LB or (ii) LB + 200 μM DIP. CFU were enumerated for the WT and the *iucA-* mutant for each biological replicate (n ≥ 6) at 1, 2, 3, 8, and 24 hours PI. Dotted lines connect WT and mutant CFU counts recovered from the same biological replicate. The Wilcoxon matched-pairs signed rank test was employed to determine statistical significance. Competitive Index (CI) values were calculated for each time point as described in the Materials and Methods. PI, post-inoculation. *$P < 0.05$, **$P < 0.01$; ns, not significant. (TIF)

**S2 Fig. Siderophore expression is upregulated under iron-limited growth conditions.** (A) qRT-PCR comparing gene expression of hvKP1 siderophores: aerobactin (*iucA*), enterobactin (*entC*), salmochelin (*iroB*), and yersiniabactin (*irp2*) in the WT and the *iucA-* strain grown in either LB + 200 μM DIP (iron-chelated media) or LB (iron-replete media) at 37°C under aerobic conditions. For qRT-PCR, each biological replicate (n = 5) was run as duplicates. Shown is the fold-change in transcription with *gyrA* used as the internal control for $2^{-\Delta\Delta C_T}$. (B) Total siderophore production was assessed on CAS agar plates (i) for the WT and the *iucA-* mutant grown in LB overnight and diluted 1:10 in PBS before plating and incubated for 16 hours at 37°C. Halos were measured and expressed as the mean area in pixels x1000. (ii). Images of representative halos (from (i)) are shown above each bar graph. Statistical significance was calculated using the Mann-Whitney $U$ tests. *$P < 0.05$, **$P < 0.01$, ***$P < 0.001$, and ****$P < 0.0001$; ns, not significant. (TIF)

**S3 Fig. Aerobactin is dispensable for adhesion and invasion under iron-rich conditions.** (A-B) Caco-2 monolayers were infected with either the WT (gray) or the *iucA-* strain (blue) grown in LB (non-chelated media) to mid-log phase ($OD_{600}$ ~ 0.5) at 37°C under aerobic conditions to test for adhesion (A) or invasion (B) at an MOI of 50. (C) Percent Invasion of Caco-2 monolayers infected with either the WT (gray) or the *iucA-* mutant (blue) grown in LB + 200 μM DIP (iron-chelated media) at an MOI of 50 normalized to adhesion. (D-E) Caco-2 monolayers were infected at an MOI of 50 with either the hvKP2 (green) or hvKP94 (blue) along with their respective *iucA-* mutants to test for adhesion (D) and invasion (E). Strains were grown in LB + 200 μM DIP (iron-chelated media) to mid-log phase ($OD_{600}$ ~ 0.5) at 37°C under

aerobic conditions, prior to infection. (F-G) Caco-2 monolayers were infected with either the WT containing the empty pFUS2 vector (gray), *iucA-* with empty vector (blue), or *iucA-*+pFUS2[*iucA-D*] complement plasmid (p[*iucA-D*]) (pink) grown in LB+200 µM DIP to mid-log phase ($OD_{600}$~0.5) at 37°C under aerobic conditions and tested for adhesion (F) and invasion (G) at an MOI of 50. (A-G) The results from 3 independent assays, each with ≥3 replicates are shown, and the data is expressed as % to WT with ±SEM. Statistical significance was calculated using the Mann-Whitney *U* test when comparing 2 groups or Kruskal-Wallis followed by post-hoc Dunn's test of multiple comparisons when comparing 3 groups. (H) Total siderophore production was assessed on CAS agar plates for either the WT containing the empty pFUS2 vector (gray), *iucA-* with empty vector (blue), or *iucA-*+pFUS2[*iucA-D*] complement plasmid (p[*iucA-D*]) (pink). Each strain was grown overnight in LB supplemented with gentamicin (10 µg/mL) and diluted 1:10 in PBS before plating. Halos were measured and expressed as the mean area in pixels x1000. Shown are the averages for each experiment (n=3). Statistical significance was determined using a one-way ANOVA followed by Tukey's post-hoc test. *$P<0.05$, **$P<0.01$, ***$P<0.001$, and ****$P<0.0001$; ns, not significant.
(TIF)

**S4 Fig. Aerobactin expression restores *iucA-* defect in translocation.** (A-B) Polarized Caco-2 cells were grown on transwell inserts and allowed to differentiate for 19 days before infecting with either the WT containing the empty pFUS2 vector (gray), *iucA-* with empty vector (blue), or *iucA-*+pFUS2[*iucA-D*] complement plasmid (p[*iucA-D*]) (pink) grown in LB+200 µM DIP+5µg/mL gentamicin to mid-log phase ($OD_{600}$~0.5) at 37°C under aerobic conditions and tested for translocation. The top and bottom chambers were sampled 30 min PI (A) and 2 hours PI (B) to measure for translocation with data expressed as percent to WT. Shown are the results from 3 independent assays (n=4) and a Kruskal-Wallis followed by post-hoc Dunn's test of multiple comparisons was performed for statistical significance. (C-D) Contingency plots (3x2) displaying the translocation outcome for each strain at 30 min PI (C) and 2 hours PI (D). The Fisher's exact test was performed for statistical significance and the resulting p-values are indicated. (F-G) C57BL6/J mice were colonized with WT, *iucA-*, or given 2% sucrose PBS-vehicle control (uninfected) (n≥2 per group). Murine proximal colons were collected 24h PI. (F) Representative images for each group of the H&E stained proximal colons (5 µm sections). (G). Blinded histopathology scoring of H&E stained samples for epithelial damage (i) and goblet cell depletion (ii).PI, post-inoculation. **$P<0.01$, ***$P<0.001$, and ****$P<0.0001$; ns, not significant.
(TIF)

**S5 Fig. Aerobactin does not influence capsule levels and fimbriae gene expression in iron-limited conditions.** (A) qRT-PCR comparing gene expression of the type 1 (*fimA* and *fimI*) and type 3 (*mrkA* and *mrkB*) fimbriae genes in the WT and the *iucA-* strain grown in LB+200 µM DIP and LB. Each biological replicate (n=3) was run as duplicates. Shown is the fold-change in transcription with *gyrA* as the internal control for $2^{-\Delta\Delta C_T}$ analysis. (B) HMV comparison via low-speed spin-down assay of the hvKP2 (green) and hvKP94 (blue) to its respective *iucA-* isogenic mutant grown to mid-log phase ($OD_{600}$~0.5) in LB+200 µM DIP (iron-chelated media), and expressed as the percent to WT. Shown is the average of 5 independent assays each performed in triplicate. Mann-Whitney *U* tests were performed to determine statistical significance. (C) Uronic acid content was quantified as a metric of capsule production for the WT (gray) and the *iucA-* mutant (blue) grown to mid-log phase ($OD_{600}$~0.5) in iron-chelated media (LB+200 µM DIP) at 37°C under aerobic conditions. Mann-Whitney *U* tests were performed to determine statistical relevance. Shown are the averages of 4 independent assays, each with 3 replicates. (D) HMV comparison of the WT containing the pFus2 empty vector (gray), *iucA-*+empty vector (blue), and *iucA-*+p[*iucA-D*] (pink) grown to mid-log phase ($OD_{600}$~0.5) in LB+200 µM DIP at 37°C under aerobic conditions and expressed as percent to WT. The results of 3 independent assays, each with 3 replicates are shown and a Kruskal-Wallis followed by post-hoc Dunn's test of multiple comparisons was performed to determine statistical significance. *$P<0.05$, **$P<0.01$, ***$P<0.001$, and ****$P<0.0001$; ns, not significant.
(TIF)

**S1 Table. Strains used in the study.**
(DOCX)

**S2 Table. Plasmids used in the study.**
(DOCX)

**S3 Table. Primers used in the study.**
(DOCX)

**S4 Table. Data used to generate figures.**
(XLSX)

## Acknowledgments

We thank Thomas Russo, MD (University of Buffalo-SUNY) for the hvKP strains and the pFUS2 plasmids used in this study. We also thank Virginia Miller, PhD (UNC-Chapel Hill) and Kimberly Walker, PhD (UNC-Chapel Hill) for valuable conversations and input on the project. We are grateful to David Ornelles, PhD (Wake Forest School of Medicine) for providing assistance with statistical analyses. We thank Gabriela De la Cruz and Yongjuan Xia in the Pathology Services Core (PSC, UNC-Chapel Hill) for expert technical assistance with Histopathology and Digital Pathology.

## Author contributions

**Conceptualization:** M. Ammar Zafar.

**Data curation:** Giovanna E. Hernandez, Md. Maidul Islam, Suhrid Maiti.

**Formal analysis:** Giovanna E. Hernandez, Md. Maidul Islam, Suhrid Maiti, David L. Caudell, Edison T. Floyd, Hannah M. Atkins.

**Funding acquisition:** M. Ammar Zafar.

**Investigation:** Giovanna E. Hernandez, Juan D. Valencia-Bacca, Emma F. Bennett, Md. Maidul Islam, Suhrid Maiti, Noah A. Nutter, Taylor M. Young, Alicia Costa-Terryll, Jaden J. Skelly.

**Methodology:** Giovanna E. Hernandez, Juan D. Valencia-Bacca, Emma F. Bennett, Noah A. Nutter, Taylor M. Young, Alicia Costa-Terryll, Jaden J. Skelly.

**Project administration:** Giovanna E. Hernandez, M. Ammar Zafar.

**Supervision:** Md. Maidul Islam, M. Ammar Zafar.

**Validation:** Md. Maidul Islam, Suhrid Maiti.

**Visualization:** Giovanna E. Hernandez, Md. Maidul Islam, Suhrid Maiti.

**Writing – original draft:** Giovanna E. Hernandez, M. Ammar Zafar.

**Writing – review & editing:** Giovanna E. Hernandez, M. Ammar Zafar.

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
