## [Decision Letter · Decision Letter 0]

31 Aug 2025

Aerobactin is a key driver of hypervirulent *Klebsiella pneumoniae* translocation and virulence translocation and virulence translocation and virulence translocation and virulence

PLOS Pathogens

Dear Dr. Zafar,

Thank you for submitting your manuscript to PLOS Pathogens. After careful consideration, we feel that it has merit but does not fully meet PLOS Pathogens's publication criteria as it currently stands. Therefore, we invite you to submit a revised version of the manuscript that addresses the points raised during the review process.

Points raised by multiple reviewers and to be addressed in a revised manuscript are genetic complementation studies and epithelial cell translocation assays. Please submit your revised manuscript within 60 days Oct 30 2025 11:59PM. If you will need more time than this to complete your revisions, please reply to this message or contact the journal office at plospathogens@plos.org. Please include the following items when submitting your revised manuscript:

We look forward to receiving your revised manuscript.

Kind regards,

Leigh Knodler

Academic Editor

PLOS Pathogens

Thomas Guillard

Section Editor

Editor-in-Chief

PLOS Pathogens

orcid.org/0000-0003-2946-9497

Editor-in-Chief

PLOS Pathogens

orcid.org/0000-0002-7699-2064

**Journal Requirements:**

- ® on pages: 19, 23, and 39

- TM on pages: 20, 21, 22, and 23.

Potential Copyright Issues:

i) Figures 1B, and 2A. Please confirm whether you drew the images / clip-art within the figure panels by hand. If you did not draw the images, please provide (a) a link to the source of the images or icons and their license / terms of use; or (b) written permission from the copyright holder to publish the images or icons under our CC BY 4.0 license. Alternatively, you may replace the images with open source alternatives. See these open source resources you may use to replace images / clip-art:

6) We note that your Data Availability Statement is currently as follows: "All data supporting the findings of this study are available within the manuscript and its supporting information files.". Please confirm at this time whether or not your submission contains all raw data required to replicate the results of your study. Authors must share the “minimal data set” for their submission. PLOS defines the minimal data set to consist of the data required to replicate all study findings reported in the article, as well as related metadata and methods (https://journals.plos.org/plosone/s/data-availability#loc-minimal-data-set-definition).

7) Please amend your detailed Financial Disclosure statement. This is published with the article. It must therefore be completed in full sentences and contain the exact wording you wish to be published.

**Reviewers' Comments:**

Reviewer's Responses to Questions

**Part I - Summary**

Reviewer #1: In this manuscript by Hernandez and colleagues, the authors seek to understand the role of aerobactin in the colonization and translocation of hypervirulent Klebsiella pneumoniae strains from the gut to sterile sites. Using in vitro and in vivo approaches, the authors define that aerobactin is responsible for Kp translocation in the gut niche to sterile sites and for overall Kp pathogenicity during infection. The experiments are well controlled, analyzed, and interpreted. My main concern is the lack of complementation and contextualization within the Kp pathogenesis field. The authors do not cite literature that is relevant to the study, which lessens the overall impact of the work. There is also a lack of in vivo complementation data that would support overall findings from this study.

Reviewer #2: The paper addresses a key ‘sink’ for acquired drug resistance, hypervirulent Klebsiella pneumoniae (hvKp), and its ability to move from the gut environment to the bloodstream where it can causes sepsis. hvKp present something of enigma insofar as they need to adhere to especially epithelial mucosae, yet often express a mucoid capsule which presents a barrier to adhesion. Once through the epithelial barrier and the iron-rich environment of the gut, they need to acquire iron where much of this key micronutrient is sequestered by host iron-binding proteins. hvKp can have multiple systems for acquiring iron, and express up to 4 siderophore systems. The authors have investigated the link between hvKp virulence, the virulence plasmid expression of aerobactin (iuc genes), gut colonisation in competition with a rich microbiome and specifically translocation through the epithelia. The authors acknowledge that considerable work has been published on closely related studies that used different models for lung and soft tissue infection, and sepsis. The authors make some interesting and important observations that adds to our understanding of siderophores in hvKp.

Reviewer #3: Siderophores are important for gut colonization by enteric pathogens, and aerobactin, a siderophore encoded on virulence plasmids commonly found in hypervirulent and convergent K. pneumoniae, has been associated with increased virulence. However, supporting evidence has been largely correlative. This study provides compelling evidence that aerobactin is not essential for gut colonization but plays a critical role in translocation of K. pneumoniae to extraintestinal sites. This finding significantly advances our understanding of K. pneumoniae pathogenesis and may influence future strategies for screening patients colonized with aerobactin-positive hvKp strains. The manuscript is clearly written, well-controlled, and will be of broad interest to the field. I offer a few minor suggestions below to enhance clarity for a general audience.

Reviewer #4: In this manuscript entitled "Aerobactin is a key driver of hypervirulent Klebsiella pneumoniae translocation and virulence" by Hernandez et al., the authors investigated the role of aerobactin in facilitating gastrointestinal translocation and systemic dissemination of hypervirulent Klebsiella pneumoniae (hvKP). Using a murine gastrointestinal colonization model with intact microbiota, the authors compared pathogenesis of an hvKP clinical isolate (hvKP1) to an isogenic aerobactin biosynthesis mutant (iucA-). It has been shown previously that in pneumonia, IP or SQ systemic infection models aerobactin plays a major role in virulence for strain hvKP1. In this work the authors are evaluating the role of aerobactin in pathogenesis in a gastrointestinal model and are challenged with discerning the role of aerobactin in the GI tract vs its already established role during a systemic infection.

The study demonstrates that while both strains colonized the GI tract similarly, mice colonized with the iucA- mutant exhibited significantly lower bacterial burdens in extra-intestinal organs and a decrease in mortality. In addition, the authors provided data suggesting that within a systemic infection model (tail vein) iuc- mutants also have a defect in virulence. They provided a variety of cell culture assays that showed reduced adhesion, invasion, and translocation capabilities of the iucA- mutant, with increased hypermucoviscosity (HMV) linked to upregulation of the rmp locus. Overall, this manuscript is very well written, with some strong evidence, iron-homestasis plays a role in regulating surface associated factors and provides substantial data to support a conclusion that iron-homeostasis is complicated and highly important for virulence in K. pneumoniae. This manuscript includes a substantial body of work that is both informative and impactful to the field.

However, many of the authors conclusions are broadly stated where the data does not support them and should be softened to allow for alternate interpretations and more accurate descriptions. The authors' conclusion that this represents a specific translocation defect is questionable, as the in vitro translocation data presented is relatively weak and would benefit from more rigorous experimental validation. While this work attempts to identify a role for aerobactin in bacterial translocation, the experimental evidence supporting translocation-specific effects is insufficient, and the interpretation of comparative extra-intestinal bacterial burden and virulence between GI and systemic models requires more thorough analysis and discussion. The authors do not provide convincing data to support their hypothesis for a role for aerobactin in translocation vs a role for aerobactin in survival and expansion in the blood stream and extraintestinal organs.

**Part II – Major Issues: Key Experiments Required for Acceptance**

Please use this section to detail the key new experiments or modifications of existing experiments that should be absolutely required to validate study conclusions.required to validate study conclusions.required to validate study conclusions.required to validate study conclusions.

Reviewer #1: - In Figure S1, I am concerned that the iucA mutant is not actually defective in iron depleted media. There are no competitive defects in direct 1:1 competitions (Figure S1 Bi and Bii) and the CFU counts for Aii and Aiii are within an error range for serial quantification. Due to the data in subsequent figures, I do not think this lessens the impact of the overall story but these data should be reframed to say that mutations in iucA do not have a dramatic effect on growth in iron depleted media but do impact dissemination. Because other siderophores are likely compensating in the absence of iucA and because icuA is impacting hypermucoviscosity, it is likely that pathogenesis effects are due to capsule and not iron uptake. As such, this work may give new insight in to a new role of aerobactin, unrelated to iron uptake, which is still interesting.

- Because iucA mutations may be affecting multiple cellular processes, it is important to complement this phenotype in vivo. The authors should demonstrate that the fitness defects observed in Figure 1E, 1G, and 2D are related to IucA itself and not off target effects.

- The paper does not appropriately contextualize these findings within the field. Previous literature has shown that most gut isolates of Kp are classical and not hypervirulent which should be mentioned. Previous literature has also shown that Kp burden at other primary cites influences dissemination to secondary sites (PMID39824859). A recent TnSeq study showed that iron uptake systems, such as tonB, are dispensable for hypervirulent Kp pathogenesis during bacteremia (PMID37463183). The findings of Figures 1 and 2 should be placed within the context of this study. Lastly, work into capsule has shown interactions between changes in hypermucoviscosity and Kp attachment and invasion (PMID40595687; PMID37610214).

Reviewer #2: The authors have made an important observation about the cell biology concerning an aerobactin synthesis mutant of hvKp – mainly that that can’t egress from an infected cell.

This is very interesting and likely important. The apparent redundancy in the KpSC siderophore systems doesn’t appear to impact here and this phenomenon warrants at least a simple microscopy study to examine where in the cells the bacteria reside, and where they are getting ‘stuck’. The key figure is Fig.4C. Using Transwell studies, the authors note that while approximately half the number of iucA- bacteria enter cells, none emerge on the ? basolateral side of the Caco-2 monolayer. This should be supported by microscopy showing differences in cell signalling. There should be plenty of resources to study eg. P13K/AKT. Whatever the cause and effect, this is not reflected by TEER, in gross histology or buy occluding H-scores.

Figure 4 – the data as presented does not as compelling as it might be. Why not represent 3-5 replicates X 3 individually as individual sample ‘dots’? It is hard to understand what level of adherence is actually observed. The MOI of 50 is high and centrifugation aids contact. In the methods, the numbers supposedly link to the inoculum (100%?) but the axis suggest it is relative to WT. Could the numbers of each of the wells tell the same story – there should be at least 10 data points for each figure (3-5 replicates X3). Since Adhesion likely precedes gentamicin resistance (i.e. Invasion), a deficiency in adhesion (4A) is likely to lead to a deficiency in invasion (4B), meaning that the deficit is in adhesion. Again 4B could be presented as raw numbers – how many of the 50 X5 X10(5) bacteria are recovered in the gentamicin protection assay.

Reviewer #3: none noted

Reviewer #4: Major Comments:

• The authors' interpretation of their systemic infection model results in the context of their gastrointestinal model is insufficient. While no discernible differences were observed between wild-type and iucA- strain bacterial burden in systemic sites at 6- or 24-hours post infection, the iucA- mutant shows a virulence defect at 8 days post infection. This comparison is not appropriate and does not support their conclusion that “These results likely preclude early clearance of the iucA- as the reason for reduced burden in the extra-intestinal sites” for the reasons described below:

o In the tail vein model, the entire 106 CFU inoculum immediately enters the bloodstream and accumulates in filtration organs, which is fundamentally different from the presumably low numbers of bacteria that translocate across the intestinal epithelium over the course of several hours in the GI model. This is evident at both 6 and 24 hours in the systemic model, where the WT and mutant had 107 CFU/gram in the spleen while in the GI model, the WT had 105 CFU/gram at 24 hours, and the mutant had 103 CFU. It is possible that the differences in CFU found in extra-intestinal organs are due to low levels of translocation into the blood stream for both WT and iucA- mutant but the WT is able to survive and expand at higher numbers than the mutant in the blood stream or extra-intestinal organs. This would not be observed in the tail vein model with administration of such a large dose. This possibility seems especially likely because the iucA- mutant has a growth defect in iron-depleted media.

o The virulence defect of the iucA- mutant in the systemic model, while smaller, is nearly as substantial as that observed in the GI colonization model. The authors fail to adequately discuss comparisons between the two models or provide compelling evidence that the virulence defect observed in the GI model is not due to impaired iron acquisition and subsequent poor survival or inadequate expansion in the extra-intestinal space as is hypothesized for the systemic model.

o Since the authors provide evidence of changes to the cell surface (which could affect survival) in the iuc- mutant, their argument would be strengthened by evaluating survival in the following experiments:

Measuring bacterial survival of WT and iucA- strains in human complement serum.

Performing a macrophage uptake/survival assay

o Taken together, these additional experiments could help distinguish a role in translocation from survival as the cause of differences in bacterial burden observed in the extra-intestinal sites.

• The authors attempted to validate their findings across genetically diverse hvKP isolates by testing hvKP2 (ST23, KL1) and hvKP94 (ST23, KL1) along with their respective aerobactin-deficient mutants. However, conclusions related to aerobactin's impacts on virulence across genetically distinct isolates are difficult to interpret for several reasons. First, genetic relatedness between the strains is not evaluated, making it unclear whether observed differences reflect strain-specific effects rather than conserved mechanisms. Second, aerobactin deletions in hvKP2 are known to have little to no defect in virulence in subcutaneous or pneumonia models, and thus the non-statistically significant differences found in the GI model are not convincing of an essential role for aerobactin in hvKP2. (DOI: 10.1016/j.ebiom.2024.105302 and unpublished data). Third, results related to hvKP94 are particularly difficult to interpret, as Russo et al. previously published this strain as a partially virulent isolate that produced very little or no siderophores under iron-limiting conditions (https://doi.org/10.1128/msphere.00045-21). Given this background, the expectation would be that an iucA- mutant would have no effect on pathogenesis since this strain is not producing meaningful amounts of siderophores to begin with. These flaws and overall lack of discussion of these discrepancies significantly undermine the authors' claims about the conservation of aerobactin-mediated effects across different hvKP genetic backgrounds. It remains clear that there is heterogeneity among hvKP isolates and the requirement of aerobactin for high levels of virulence is limited in scope. Conclusions could be strengthened by testing these strains and their corresponding iucA- mutants in some of the in vitro assays such as CAS, hmv, adhesion, invasion, and translocation.

• The authors' conclusions about translocation in the GI model rely heavily on an in vitro Caco-2 model that demonstrates a translocation defect at only a single timepoint (30 minutes) with highly variable results. While TEER measurements were extended to 6 hours post-infection, the translocation assay itself was limited to a single early timepoint. These conclusions would be significantly strengthened by:

o Including multiple timepoints for translocation assessment

o Testing the complemented strain to confirm the specificity of the observed defect

o Additionally, to strengthen their conclusion that aerobactin contributes to translocation across distinct hvKP lineages, the authors would benefit from performing additional transwell assays using the hvKP2 and hvKP94 strains and their respective mutants

• The authors suggested increased expression of rmpD may be causing the observed defect in adhesion and thus poor translocation of the iucA- strain. Because the rmpD- and ΔiucA rmpD- strains have high levels of adhesion, then including these strains in the GI model or transwell experiments would clarify if the phenotypes caused by the elevated expression of rmpD is in fact contributing to a translocation defect. i.e., rmpD mutant adheres better does this improve translocation?

**Part III – Minor Issues: Editorial and Data Presentation Modifications**

Reviewer #1: - Figure S2 Bi should more clearly indicate which strains are spotted on the plate in which position.

- Can the authors elongate the x-axis in Figure 1F? The data that is difficult to read due to variability and close proximity. The authors could also consider displaying data as a box plot or a violin plot to better visualize grouped data.

- Figure 2. Can the authors speculate on why mice would have a lower survival rate in the presence of aerobactin but no differences in CFU?

Reviewer #2: There is considerable redundancy in the iron acquisition systems of Kp, especially in vitro where high potency iron chelators are deployed to sequester any available Fe to test ‘essentiality’ for growth. Some attempts have been made to establish a hierarchy for these systems but their apparently redundant existence points to evolutionary drivers that are complex and which may sit outside of the human host. The broader Kp Species Complex (KpSC) is dominated by clinical isolates and the sequence database and strain collections are dominated by isolates from human disease. This may just be the ‘tip of the iceberg’ where this important family of bacteria are concerned.

Finding a growth defect in vitro for a iucA mutant, when the siderophore synthesis gene but not the rest of the aerobactin system allows mutant complementation in WT vs iucA- competition assays. This is currently a Supp figure but maybe the key panel could be added to Figure 1. This suggests that the quantity of aerobactin produced by the WT is sufficient to ‘feed’ both strains. It would have been interesting to see whether the aerobactin mutant has a phenotype in iron-free (not depleted) minimal medium, to which increasing concentrations of Fe are added. LB is a very rich medium and a poor test of the metabolic capacity of bacteria.

Hence, Line 158 should be “aerobactin synthesis is not required for colonisation”.

Oral feeding leading to death (Fig. 1E) clearly demonstrates that the isolate is hypervirulent.

In the Fig. S2B(i) there seems to be a large difference in the siderophores produced by the iucA- mutant suggesting a lack of compensation by the remaining siderophores? This Figure could be better labelled. The transcription data in the same Supp figure is also telling – the apparent compensation by entC, iroB and irpB represented by transcript analysis (Fig. S2A), is not reflected in the CAS agar data (Fig. S2B).

In gut colonisation, is it possible that the apparent redundancy for iucA in colonisation is because only the iucA gene is deleted in this study, and that aerobactin is provided by the microbiome? Was the construction of a transporter mutant considered? The level of colonisation from a 10(6) dose without stomach acid neutralisation is another impressive demonstration of virulence. It would also be interesting to know whether the lack of ieuA impacts gut-mediated transmission in co-housing experiments; the phenotype missing from growth in the the gut lumen (Fig 1D), might be observed.

The observation regarding the lack of MLN colonisation (cf S. Typhimurium) is an interesting one, and likely correct. Salmonellae have an interesting relationship with phagocytes which may be absent in Kp, because the Kp capsule makes phagocytosis very challenging. However, there were some mice that had MLN colonisation, and others none - the effect looks binary.

Again relegating what are important and different mouse clearance kinetics of the WT and ieuA- strains to Supplementary data (S3) makes it unnecessarily tricky for the reader.

The data on the siderophore-negative strains are compelling though the interpretation that all are necessary (line 209) probably requires experimental support. It may be that aerobactin plus one other siderophore is sufficient.

The deficiency in translocation in the ieuA isolate is well supported by the systemic infection data.

While the contribution from blood is likely to be limited (given the blood data, 2B; 2C), blood-rich organs like the liver and spleen should be perfused prior to analysis of bacterial counts. Was this done but not mentioned?

The data in Figure 2C suggest that aerobactin is not essential for bacterial growth in a sepsis model, as many more bacteria were recovered than injected, and this happened in the iucA mutant too. This is counterintuitive, though it may simply reflect siderophore redundancy. Was any thought given to trying to measure aerobactin levels in the blood of the mice, uninfected or infected to determine whether the microbiome is providing sufficient aerobactin to support growth of the bacterium that is unable to synthesize aerobactin?

In Figure 3, subtle differences might be marked by too small an ‘n’. Which siderophore genes did the Kp strains in Figure 3 carry? Is there simple genomic evidence of strain diversity?

Do we know that the key entry point/pathology for hvKp is the colon?

Line 299 – wzi is thought to be involved in capsule retention, not synthesis

The study of encapsulation is made more difficult by the fact that strains with large capsules are hard to pellet and the pellet is very friable. Uronic acid after vigorous vortexing (assuming the capsule contains a uronic acid) is a direct way of measuring capsule production, were not different yet the assays for gene transcription suggested a difference. The effects of aerobactin on transcription of the genes assessed can be direct or indirect – direct is implied but the level of transcription of all the genes tested seems to be affected to a similar level (Fig 5B), or near similar level (Fig. 5C). This doesn’t spell specificity. It would be comforting that this wasn’t a general effect resulting from a higher level function (e.g. a low iron stress response) to see 2-3 housekeeping that were not impacted by the mutation. GyrA is mentioned but the Ct values or some more primary data measure should be presented, to facilitate reproduction and to see the actual levels of transcript being detected.

In Fig. 5F, it is hard to see how a 3 vs 3 comparison with considerable overlap could yield such a high P-value.

Discussion

The interpretation is clear and reasonable, until attributing cause-effect (line 394 onwards). It is well established that capsule thickness is negatively correlated with adhesion and hence the reductions in invasion, which were consistent with the reduction in adhesion, might simply result from increased expression of the capsule. The transcription analysis would be better served by an RNASeq approach, rather than selected gene analysis. An unbiased analysis would show what specific functions are affected, or whether it is a more general transcriptional response to stress caused by reduced iron.

The Figures are well presented but the data is typically highly processed in most cases and often referenced back to % of the WT. This does not readily allow a deeper understanding by the reader as relative measure are subject to changes in either the numerator or denominator.

General

There are some key data to the arguments presented currently residing in Supplementary Data, and some interesting data too that, though not germane to the core argument, provide some insights into the aerobactin mutant. The reviewer appreciates that a line must be drawn but having to access Supplementary data to support key conclusions is a frustration. Given the online journal format, it should be possible to add in some of the positive data found in the Supps to the manuscript, without incurring much in costs.

Fore example, iven the data in Fig S1A, the competition assay (Fig. S1B) is a clear demonstration that sufficient arobactin is produced to support full growth of iucA- bacteria.

The statistical analysis too has some anomalies – while the data in Fig. S5A mrkA is ‘ns’ without any significant overlap in the data, the Data for the complementation (Fig. S4.D) has significant overlap yet is strongly statistically different. Statically significant comparisons of 3 biological replicates with 3 biological replicates should not allow for any overlap of the three samples.

In the strains table, the siderophore intact genes should be listed, at least for the wild type. Is the parent strain generally available.

Reviewer #3: 1. Lines 91-95 could be expanded to include greater detail as this section is critical information for this study.

2. Line 157-159: To improve clarity for the reader, the authors could include a sentence stating that because fecal shedding reflects the number of bacteria present in the gut lumen, equivalent shedding between wild-type and mutant strains suggests that both are similarly capable of colonizing the gastrointestinal tract.

3. Figure 1H-I: Really nice data

4. Line 181-182: ‘We inoculated mice at a 1:1 ratio of the WT and the mutant strain, and as expected both strains colonized equally well in the GI tract.’ It looks like the WT outcompeted the mutant in the gastrointestinal samples (F: fecal, CE: cecum, CO: colon). Were there stats performed here?

5. Fig. S2Bi was difficult to see - more labeling needed to improve clarity. Also, more description in the figure legends would help improve clarity. Perhaps excising the halos and inserting them above the bar graph in Fig. S2Bii would help?

6. I have no issues assessing the data, but the figures are low resolution even in the downloaded pdf.

7. The data in SFig3i-ii is very important. The authors should consider moving it to a main figure.

8. Fig4A-B: really nice data

9. Line 282-283: Additionally, T3F bind to abiotic catheters and facilitate UTIs in mice (Murphy et al. Role of Klebsiella pneumoniae type 1 and type 3 fimbriae in colonizing silicone tubes implanted into the bladders of mice as a model of catheter-associated urinary tract infections. Infect Immun. 2013;81(8):3009-17).

10. Line 289-290: ‘These results suggest that the reduced epithelial cell engagement observed in the iucA- mutant is likely independent of fimbrial gene expression.’ Is it possible that the fimbriae are involved in adherence but this is not evident at the transcriptional level? If fimbriae expression is activated in the WT and iucA mutant under iron deplete conditions, but HMV is only reduced in the WT, perhaps this helps expose the fimbriae to mediate adherence. The authors could consider discussing this or testing adherence of the fimbriae mutants under iron deplete conditions.

11. Fig5E: The data here is expressed as % adherence to WT. It would be nice to see the raw adherence data (even as supplemental) to get a sense of the % of bacteria that adhere to host cells under iron deplete conditions. It would be nice to compare this to the raw data under iron replete conditions.

12. The discussion is accessible and enjoyable to read. But given how this study has highlighted the importance of aerobactin in mediating translocation from the gut, it would be useful to add a sentence to discuss if there may be a benefit to screening at risk patients for Kp strains that express aerobactin in their gut, particularly if they will be / have been administered antibiotics.

Reviewer #4: • The introduction would benefit from a more balanced discussion of aerobactin's role in hvKP virulence. The current text presents aerobactin as consistently associated with enhanced virulence, but this oversimplifies the literature. I recommend the authors acknowledge several important discrepancies:

1. Convergent isolates: Some isolates that acquire aerobactin through horizontal gene transfer do not exhibit the expected hypervirulent phenotype, suggesting aerobactin alone is insufficient for hypervirulence.

2. Aerobactin-negative hvKP: Certain hvKP strains lacking aerobactin have been documented to cause severe invasive infections, indicating heterogeneity in iron-homestasis mechansims among hvKP.

3. Retained virulence in mutants: Previous reports have described aerobactin-deficient mutants that maintain significant virulence, contradicting the notion that aerobactin is "crucial" for pathogenesis.

Minor Comments:

• Ln 114-116: “..aerobactin is not necessary for gut colonization, it is crucial for translocation..” this should be softened, the iucA- mutant does get into extraintestinal organs in ~50% of mice.

• Ln 140- 143: “a more pronounced upregulation of the other siderophores..” This part of the sentence should be modified, only salmochelin displayed a statistically significant increase. Enterobactin and yersiniabactin expression were not increased in the iucA- mutant.

• Ln 149-151 and Fig 1C: “.. samples revealed elevated levels of all four siderophores..” Are iroB and irp2 statistically significant? It may also be worth noting that these expression levels are a fraction of those in iron-depleted media. Which could be related to the challenges of in vivo qRT-PCR or could suggest that the gut while iron limiting is not as limited as iron-depleted LB (which could explain why the iucA- mutant colonizes the GI tract as well as the wild type).

• Ln 179-185: “We speculate that, compared to the gut, the differences observed in systemic sites could be due to the infection barriers present within the host.” I agree, but I think this should be expanded upon in the discussion. For example, co-colonization data in the GI model (Fig S3A) suggests a tight bottle neck and founder effect in disseminated tissues.

• Ln 277 “Aerobactin regulates the HMV phenotype in hvKP” Please be careful when describing the role of aerobactin, aerobactin does not directly “regulate” the HMV phenotype, it binds iron in the extracellular environment and brings it back to the cell. Expression of rmp is modulated by Fur. In a manuscript defining a new role for aerobactin it is important to be clear in technical writing so as not to be misinterpreted. Suggest something like “Aerobactin mutants have increased HMV”

• Ln 394-415: As stated, these cell surface factors are known to be regulated by Fur. These changes in the aerobactin mutant seem to be global and the result of iron-deprivation. Since Fur repression is mediated by the availability of iron, some discussion / speculation should be included here about why expression of other siderophores is not sufficient to provide the cell with enough iron. Previous data from Russo et al., (https://doi.org/10.1128/iai.01667-13) suggested that hvKP1 and its corresponding iucA- mutant do not produce any additional siderophores (by HPLC), despite their biosynthetic enzymes being expressed, as shown in this manuscript.

• Ln 613 and 1009: The author’s state the transwell assay in Figure 4C shows the averages of 3 independent experiments. However, there are 4 datapoints per strain. Please clarify.

• Figure images are blurry.

• Please provide raw data for all experiments as a supplemental table: Inoculum for all mouse infections, inoculum and CFU values for top and bottom layers for translocation assays, raw data for normalized figures such as Fig 4A-B and 5D-F, etc.

• Please update the methods for animal experiments to discuss randomization and blinding of experiments to comply with ARRIVE guidelines, a requirement of PLOS Pathogens.

• If mice were euthanized due to a loss of >=20% initial body weight, please provide body weight measurements as a supplemental table.

PLOS authors have the option to publish the peer review history of their article (what does this mean?). If published, this will include your full peer review and any attached files.). If published, this will include your full peer review and any attached files.). If published, this will include your full peer review and any attached files.). If published, this will include your full peer review and any attached files.

...

Reviewer #1: No

Reviewer #2: **Yes:** Richard StrugnellRichard StrugnellRichard StrugnellRichard Strugnell

Reviewer #3: No

Reviewer #4: No

**Figure resubmission:**

**Reproducibility:**



---

## [Decision Letter · Decision Letter 1]

17 Mar 2026

PPATHOGENS-D-25-01441R1

Aerobactin is a key driver of hypervirulent *Klebsiella pneumoniae* translocation and virulence translocation and virulence translocation and virulence translocation and virulence

PLOS Pathogens

Dear Dr. Zafar,

Thank you for submitting your revised manuscript to PLOS Pathogens. After careful consideration, we feel that while you have addressed many of the concerns from the reviewers, two reviewers feel that there are some still some minor issues to address, and I agree with them. We invite you to submit a revised version of the manuscript that addresses the points raised.

We look forward to receiving your revised manuscript.

Kind regards,

Leigh Knodler

Academic Editor

PLOS Pathogens

Thomas Guillard

Section Editor

PLOS Pathogens

Sumita Bhaduri-McIntosh

Editor-in-Chief

PLOS Pathogens

orcid.org/0000-0003-2946-9497

Michael Malim

Editor-in-Chief

PLOS Pathogens

orcid.org/0000-0002-7699-2064

**Journal Requirements:**

At this stage, the following Authors/Authors require contributions: Md. Maidul Islam, and Suhrid Maiti. Please ensure that the full contributions of each author are acknowledged in the "Add/Edit/Remove Authors" section of our submission form.

- TM on page: 25.

3) Thank you for including an Ethics Statement for your study. Please include:

i) A statement that formal consent was obtained (must state whether verbal/written) OR the reason consent was not obtained (e.g. anonymity). NOTE: If child participants, the statement must declare that formal consent was obtained from the parent/guardian.].

4) Please amend your detailed Financial Disclosure statement. This is published with the article. It must therefore be completed in full sentences and contain the exact wording you wish to be published.

2) If any authors received a salary from any of your funders, please state which authors and which funders..

5)  Please ensure that the funders and grant numbers match between the Financial Disclosure field and the Funding Information tab in your submission form. Note that the funders must be provided in the same order in both places as well.

**Reviewers' Comments:**

Reviewer's Responses to Questions

**Part I - Summary**

Reviewer #2: The authors have addressed many of the key issues. The attempts at complementation were unsuccessful with the gene (operon structure< regulation?) but there was some short term complementation in vivo using a plasmid carrying multiple genes. Complementation is tricky but the purpose is to lend weight to conclusions. If the authors are unable to complement the bacteria with construct that is stable in vivo, it is very hard to be dogmatic that aerobactin is the key product missing from the mutants.

The data presented in the rebuttal Figure are supportive but the loss of plasmid in vivo is not well explained. If it is simply a replicon instability issue then there are multiple replicons with strict (vs relaxed) replication controls that should be retained.

Reviewer #4: This revised manuscript prepared by Hernadez et al., is very much improved. Specifically with regards to scientific rigor and robust support of their conclusions. However, some issues remain requiring clarification of humane endpoints utilized in mouse studies, and some slight misinterpretation of the published literature which could modify the tone and narrative of the discussion.

**Part II – Major Issues: Key Experiments Required for Acceptance**

Please use this section to detail the key new experiments or modifications of existing experiments that should be absolutely required to validate study conclusions. required to validate study conclusions. required to validate study conclusions. required to validate study conclusions.

Reviewer #2: The issue around the large MOI is not well addressed – if the same MOI is used in other studies then both might be criticized. The purpose of the translocation study was to show that the bacteria can move through a cell monolayer. Ignoring the impact of centrifugation, if in vivo there are good examples where there are 50 bacteria for every cell in the epithelial barrier then such an MOI might be justified. Otherwise the studies risks observation of an artefact resulting from using very high bacterial numbers.

Out of interest, were lower numbers of bacteria ever used? Lacking classical bacterial epithelial ‘invasion’ processes like Type 3 secretion systems, Kpn will need some other mechanism for creating changes to the cell membrane seen is either invasion or paracellular egress. The suggestion of using microscopy was to ask the question whether Kpn are seen within, or between, cells.

Reviewer #4: 1. Translocation defect / serum survival: While it is true that complement is likely the first line of defense Klebsiella pneumoniae faces post translocation, the serum killing assay as performed does not support or refute the conclusion that “the decreased number of translocated bacteria in mutant are not due to host clearance mechanism at an early timepoint, but rather a defect in translocation.” While I have concerns about the serum killing assay data as presented and the experimental methods, the in vitro translocation data presented in Fig 6C and S4A-B are convincing and demonstrate a clear translocation defect in the hvKP1 iucA mutant that has been restored by expressing iucABCD on a plasmid. Discussion around the results of the serum killing assay should be tempered as a siderophore mutant is unlikely to be more sensitive to 10% rabbit serum than WT. Serum killing assays are usually done with higher concentrations of serum. For example, a previous study found that using hvKP1 ΔiucA there was a growth / survival defect in 90% human serum compared to WT (PMID 26056379). Regardless, conclusions surrounding the translocation defect are supported directly by other experiments.

2. Clarification of Humane Endpoint Criteria: The Methods state that mice were euthanized if they were unable to eat or drink, lost ≥20% of body weight, and exhibited respiratory distress. Are all three criteria necessary to justify euthanasia? The survival data in Figure 4 and corresponding weight loss data in TableS1_data show that mice were allowed to survive after losing ≥20% of body weight and in some cases for several days before being marked as a death. Because the survival curves in Figure 4 could differ substantially depending on how endpoints were applied, please clarify how humane endpoint criteria were utilized for these studies. If weight loss ≥20% of initial body weight is not a hard fast rule for euthanasia, as is typical for IACUC approved protocols, please clarify. With some strains I have observed mice losing significant weight while showing no other signs of infection. I believe that clarification on endpoint criteria should be sufficient to alleviate this concern. In future studies, utilization of a clinical scoring system by an individual blinded to the study and / or ensuring investigators are blinded to strain identity would further enhance transparency and eliminate ambiguity.

3. Discrepancies Between Survival and Weight Data: In the iucA group, Figure 4D indicates that four mice were alive at Day 8. However, the longitudinal weight table shows only three mice with recorded body weights at Day 8 (192 h). Please clarify whether one animal’s Day 8 weight measurement is missing from the table or whether there is a discrepancy between the survival and weight datasets. In addition, in the data for Figure 4D, deaths and weight loss measurements in the iucA mutant groups do not match. For example, mouse one was listed as dead on day 7 but there is weight loss data on day 8. Please clarify / resolve the discrepancies between the deaths in Figure 4D and the weight loss data.

4. Interpretation of Prior Siderophore Data (Russo et al.): The revised manuscript and accompanying responses state that, based on Russo et al. (PMID: 24664504), “aerobactin is the primary siderophore produced (90%), and the other 3 make up the rest (10%).”

This interpretation does not accurately reflect the data presented in the cited study. In Russo et al., 100% of detectable siderophore activity (by HPLC fractionation and CAS assay) was found within a single coarse HPLC fraction (fraction 5). Upon further sub-fractionation, approximately 90% of siderophore activity was localized to fraction 5c and ~10% to fraction 5d. Fraction 5c was chemically identified as aerobactin. However, fraction 5d was not structurally characterized as a distinct siderophore. The minor activity detected in fraction 5d more likely reflects an aerobactin-related species rather than a distinct siderophore system, as it co-fractionated with aerobactin and was not independently characterized. Importantly, the authors explicitly state: “As 5c and 5d elute in neighboring fractions, the activity in fraction 5d probably represents carryover from fraction 5c.”

Furthermore, in the ΔiucA strain, siderophore activity was no longer detected in fraction 5, and only minimal background activity was observed in fraction 1. Given that fraction 1 corresponds to early-eluting, highly polar, low–molecular-weight species under the reverse-phase conditions used, this weak signal is unlikely to represent production of a structurally defined siderophore such as enterobactin, salmochelin, or yersiniabactin. Rather, these data further support the conclusion that aerobactin is the sole functionally significant siderophore produced by hvKP1 under the tested conditions.

It is biologically plausible that a strain the produces high-levels of aerobactin may have its ability to produce the other siderophores diminished, even when their biosynthetic enzymes are present and expressed). Notably, yersiniabactin and enterobactin (and thus salmochelin) share the same chorismate precursor while aerobactin biosynthesis utilizes a citrate backbone.

While it is possible that the HPLC conditions utilized in the referenced study were not optimal for resolving all potential siderophores, and thus they were missed. The results of the referenced study do not suggest that “aerobactin is the primary siderophore produced (90%), and the other 3 make up the rest (10%).” Rather the data show that all detectable siderophore activity in hvKP1 co-fractionated with aerobactin under the tested conditions.

The in vivo phenotypes described in hvKP1 ΔiucA may therefore reflect deletion of the only functionally significant siderophore produced by this strain. In contrast, strains such as hvKP2, NTUH-K2044, and KPPR1 retain high virulence despite deletion or absence of iucA, suggesting strain-specific differences in functional redundancy of iron acquisition systems. This possibility should be discussed more explicitly.

**Part III – Minor Issues: Editorial and Data Presentation Modifications**

Reviewer #2: (No Response)

Reviewer #4: The data related to diverse strains of hypervirulent Klebsiella (hvKP1, hvKP2, and hvKP94) strengthen their results and conclusions. hvKP94 is described as K1-ST23 in this study and in the prior publication from which the strain was obtained. Could the authors clarify how sequence type (ST) and capsule type (K/KL) were determined for hvKP94? Specifically, was whole-genome sequencing used for assignment, or were these designations based on prior serologic or PCR-based typing?

Line 401-403: “possibly due to expression of colibactin..” To me this seems unlikely, loss of iucA would be expected to impair iron scavenging under-host restricted conditions. Colibactin is a genotoxin and would not be expected to compensate for siderophore-mediated iron uptake. In addition, a previous study found that in NTUH-K2044 (which does not contain colibactin) an iucA deletion had no discernable effect on virulence albeit in a different animal model (PMID: 18433330). Is it more likely that while iuc is the primary siderophore produced in nearly all hvKP isolates, hvKP2 is better equipped to utilize its secondary siderophore systems than hvKP1?

PLOS authors have the option to publish the peer review history of their article (what does this mean?). If published, this will include your full peer review and any attached files.). If published, this will include your full peer review and any attached files.). If published, this will include your full peer review and any attached files.). If published, this will include your full peer review and any attached files.

**Do you want your identity to be public for this peer review?** For information about this choice, including consent withdrawal, please see our  For information about this choice, including consent withdrawal, please see our  For information about this choice, including consent withdrawal, please see our  For information about this choice, including consent withdrawal, please see our Privacy Policy....

Reviewer #2: **Yes:** Richard StrugnellRichard StrugnellRichard StrugnellRichard Strugnell

Reviewer #4: No

**Figure resubmission:**
---

## [Editor Report · Decision Letter 2]

26 Mar 2026

Dear Dr. Zafar,

We are pleased to inform you that your manuscript 'Aerobactin is a key driver of hypervirulent *Klebsiella pneumoniae* translocation and virulence' has been provisionally accepted for publication in PLOS Pathogens. translocation and virulence' has been provisionally accepted for publication in PLOS Pathogens. translocation and virulence' has been provisionally accepted for publication in PLOS Pathogens. translocation and virulence' has been provisionally accepted for publication in PLOS Pathogens.

Best regards,

Leigh Knodler

Academic Editor

PLOS Pathogens

Thomas Guillard

Section Editor

PLOS Pathogens

Sumita Bhaduri-McIntosh

Editor-in-Chief

PLOS Pathogens

orcid.org/0000-0003-2946-9497

Michael Malim

Editor-in-Chief

PLOS Pathogens

orcid.org/0000-0002-7699-2064
---

## [Editor Report · Acceptance letter]

Dear Dr. Zafar,

We are delighted to inform you that your manuscript, "Aerobactin is a key driver of hypervirulent *Klebsiella pneumoniae* translocation and virulence," has been formally accepted for publication in PLOS Pathogens. translocation and virulence," has been formally accepted for publication in PLOS Pathogens. translocation and virulence," has been formally accepted for publication in PLOS Pathogens. translocation and virulence," has been formally accepted for publication in PLOS Pathogens.

Best regards,

Sumita Bhaduri-McIntosh

Editor-in-Chief

PLOS Pathogens

orcid.org/0000-0003-2946-9497

Michael Malim

Editor-in-Chief

PLOS Pathogens

orcid.org/0000-0002-7699-2064